



# Photochemical Organonitrate Formation in Wet Aerosols

Yong Bin Lim[1,2], Hwajin Kim[1,3], Jin Young Kim[1,3], and Barbara J. Turpin[4]

[1]Center for Environment, Health and Welfare Research, Korea Institute of Science and Technology, Seoul 02792, Republic of Korea

[2]Department of Environmental Sciences, Rutgers University, New Brunswick, New Jersey 08901, USA

[3]Department of Energy and Environmental Engineering, University of Science and Technology, Daejeon 34113, Republic of Korea

[4]Department of Environmental Science and Engineering, University of North Carolina, Chapel Hill, North Carolina 27599, USA

*Correspondence to*: Yong Bin Lim (ylim@kist.re.kr) and Jin Young Kim (jykim@kist.re.kr)

**Abstract.** Water is the most abundant component of atmospheric fine aerosol. However, despite rapid progress, multiphase chemistry involving wet aerosols is still poorly understood. In this work, we report results from smog chamber photooxidation of glyoxal and OH – containing ammonium sulfate or sulfuric acid particles in the presence of $NO_x$ and $O_3$ at high and low relative humidity. Particles were analyzed using ultra high performance

liquid chromatography coupled to quadrupole time-of-flight mass spectrometry (UPLC-Q-TOF-MS).

During the 3 hour irradiation, OH oxidation products of glyoxal that are also produced in dilute aqueous solutions (e.g., oxalic acids and tartaric acids) were formed in both ammonium sulfate (AS) aerosols and sulfuric acid (SA) aerosols. However, the major products were organonitrogens (CHNO), organosulfates (CHOS), and organonitrogen-sulfates (CHNOS). These were also the dominant products formed in the dark chamber indicating

non-radical formation. In the humid chamber (> 70 % RH), two main products for both AS and SA aerosols were organonitrates, which appeared at m/z[-] 147 and 226. They were formed in the aqueous phase via non-radical reactions of glyoxal and nitric acid, and their formation was enhanced by photochemistry because of the photochemical formation of nitric acid via reactions of peroxy radicals, $NO_x$ and OH during the irradiation.

## 1   Introduction

Atmospheric aerosols affect human health and climate (Seinfeld and Pandis, 1998), and a large fraction is secondary organic aerosol (SOA) (Zhang et al., 2007). SOA forms via partitioning of semi-volatile organic aerosols from gas-phase oxidation of volatile organic compounds (VOCs) (Odum et al., 1996; Pankow, 1994), multiphase reactions involving aerosols (Jang et al., 2002; Kalberer et al., 2004), and aqueous-phase reactions in cloud/fog waters (Blando and Turpin, 2000). SOA formation via aqueous chemistry (SOA$_{aq}$) has been suggested

by recent laboratory, field and modeling studies (El-Sayed et al., 2015; Ervens et al., 2011; Gong et al., 2011; Lee et al., 2012; Lee et al., 2011; Lin et al., 2012; Liu et al., 2012a; Liu et al., 2012b; McNeill et al., 2012; Myriokefalitakis et al., 2011; Ortiz-Montalvo et al., 2014; Ortiz-Montalvo et al., 2012; Tan et al., 2012; Zhou et al., 2011). Considering SOA$_{aq}$ is likely to improve model predictions, which currently underestimate ambient measurements and the oxidation state of organic aerosols, because water soluble organic compounds with a small

carbon number (C2-C3), which were not considered by the partitioning theory (Pankow, 1994) due to high vapor



pressure, are precursors for SOA$_{aq}$. In addition, these water soluble organics and liquid water are abundant in the atmosphere (Blando and Turpin, 2000; Carlton and Turpin, 2013; Liao and Seinfeld, 2005).

SOA$_{aq}$ formation is expected in cloud/fog droplets and aerosol waters via radical and non-radical reactions (Lim et al., 2010; McNeill et al., 2012). Since OH radicals with the concentrations of $10^{-14}$—$10^{-12}$ M are available in
the atmospheric aqueous phase (Arakaki et al., 2013; Ervens et al., 2014), water soluble organic compounds (e.g., glyoxal, methylglyoxal) undergo photooxidation forming dicarboxylic acids (e.g., oxalic acids) via peroxy radical reactions and oligomers via organic radical-radical reactions (Altieri et al., 2008; Lim et al., 2010; Lim et al., 2013; Tan et al., 2012; Tan et al., 2009). Even without OH radicals, the photolysis of pyruvic acid also forms oligomers in concentrated solutions (Guzman et al., 2006). Organic compounds form organosulfates with sulfuric acids via
acid catalysis (i.e., with acidic sulfate) (Surratt et al., 2010; Surratt et al., 2008; Surratt et al., 2007a). Water soluble organic compounds like glyoxal and methylglyoxal form oligomers through hemiacetal formation and aldol condensation, especially in evaporating droplets (Loeffler et al., 2006; Noziere et al., 2009; Schwier et al., 2010). Ammonium ions, which are abundant in atmospheric aerosols (Zhang et al., 2007), form nitrogen-containing organics including imidazoles with water soluble organic compounds (Galloway et al., 2009; Kampf et al., 2012;
Noziere and Cordova, 2008), and also act as catalysts enhancing oligomer formation (Nguyen et al., 2014; Noziere et al., 2009). Water soluble carboxylic acids (Blando and Turpin, 2000; Chebbi and Carlier, 1996) and cations (e.g., Na$^+$, K$^+$, NH$_4^+$, protonated amines) form low volatility carboxylate salts (Ortiz-Montalvo et al., 2014; Paciga et al., 2014; Smith et al., 2010).

Although these findings have significantly improved our understanding of aqueous chemistry, atmospheric
processes like gas-wet particle partitioning of water soluble organic compounds, OH radicals and water in aerosols are still poorly understood. Therefore, organic chemistry in the presence of wet aerosol warrants further study. Studies of wet aerosol chemistry to date have demonstrated that the chemistry in wet aerosols can be quite different than that under dilute (cloud-relevant) conditions. For example, in reaction vessel experiments for photooxidation of glyoxal or methylglyoxal in the dilute aqueous phase, sulfuric acid does not affect the formation of major
products (e.g., oxalic acid) (Tan et al., 2009), while both smog chamber and field studies observe organosulfate formation in aerosols in the presence of sulfuric acid, which contributes both liquid water and acidity to condensed phase aerosol chemistry (Galloway et al., 2009; Surratt et al., 2007a; Surratt et al., 2007b; Tolocka and Turpin, 2012) . In fact, several key questions remain to be answered. For example, the major sink of nitrate radicals in the aqueous phase is the formation of HNO$_3$ (Kirkland et al., 2013). However, it should be noted that organonitrates
are commonly observed in rain waters (Altieri et al., 2009), clouds (Boone et al., 2015) and wet aerosols (Hodas et al., 2014). Are organonitrates formed in the aqueous phase? Are they taken up into atmospheric waters after gas phase formation during the daytime (ROO· + NO → RONO$_2$) or the nighttime (R + NO$_3$ → RONO$_2$)? Or are they formed via aqueous chemistry?

A smog chamber is ideal to explore condensed-phase chemistry to address these issues. Volkamer et al. (2009)
conducted smog chamber experiments for aqueous chemistry of glyoxal in wet particles. In their experiments, glyoxal was photochemically produced from the ethene-OH reaction in the gas phase, and partitioned into wet seed particles (e.g., ammonium sulfate, ammonium bisulfate, fulvic acid) with the RH range from ~ 20 % to ~ 60 %. Clearly, glyoxal is volatile, not semivolatile. Nevertheless, it forms SOA due to the high water solubility. This provided evidence that SOA yields were correlated with the water content, not the organic mass portion in



aerosols. Then, Galloway et al. (2009) studied glyoxal uptake by ammonium sulfate particles in a smog chamber, conducting dark and irradiated experiments at ~ 60 - 70 % RH. While Volkamer et al. focused on SOA yields, Galloway et al. characterized aerosol products from dark/irradiated reactions using a high resolution time-of-flight aerosol mass spectrometer, identifying imidazoles among other organic nitrogen products from dark reactions and

glycolic acid sulfates (C2H3O6S1) among other organosulfate products from irradiated reactions (Note that Volkamer et al. conducted OH radical chemistry, but Galloway et al. had no OH source). Later, Kampf et al. (2012) studied further glyoxal-ammonium sulfate aerosols identifying various imidazoles and oligomers under dark reactions using a high performance liquid chromatography coupled with a tandem mass spectrometer. Chamber studies for isoprene epoxides (IEPOX) in the aqueous phase have been also conducted. Surratt et al.

(2010) observed in a dark chamber (~ 30 % RH) IEPOX taken up by acidic sulfate aerosols formed oligomers presumably in the aqueous phase since sulfuric acid takes up water even at low RH. Nguyen et al. (2014) observed that liquid water content strongly correlated with IEPOX uptake and oligomer formation by ammonium sulfate seed particles in the presence of aerosol liquid water.

There are still few smog chamber studies addressing condensed-phase chemistry explicitly in terms of radical

reactions (irradiated reactions) and non-radical reactions (dark reactions); these two types of aqueous chemistry must be explored to understand $SOA_{aq}$ formation in aerosols. One of the challenges in studies of SOA formation through wet aerosol chemistry is that the concentration of aerosol water, the medium of the aqueous-phase reactions, depends on the 1) hygroscopicity of aerosols, 2) concentration of aerosol particles and 3) RH. Furthermore, product formation also depends on 1) the gas-phase transport of water soluble organic compounds,

2) OH partitioning between the gas and aqueous phases, and 3) competition in the aqueous phase between photooxidation and dark (non-radical) reactions.

In this work, we conduct multiphase photochemical oxidation and dark reactions in the presence of wet aerosols: the glyoxal-$H_2O_2$-ammonium sulfate aerosols (AS aerosols) and glyoxal-$H_2O_2$-sulfuric acid aerosols (SA aerosols) in a smog chamber under low (< 5 % RH) and high (> 70 % RH) humidity conditions. Ammonium sulfate and

sulfuric acid were used for seed particles to observe ammonium interactions, liquid water and acidity effects. $H_2O_2$ provided a source of OH radicals in the wet aerosols during the irradiation. $NO_x$ and $O_3$ were initially introduced into the chamber with modest concentrations ($[NO_x]_{initial}$ = 3—83 ppb, $[O_3]_{initial}$ = 0—12 ppb) to reflect chemistry under anthropogenic influences (Carlton and Turpin, 2013; Ervens et al., 2011). After chamber reactions wet aerosols were characterized by ultra-performance liquid chromatography quadrupole time-of-flight mass

spectrometry (UPLC-Q-TOF-MS).

## 2    Experimental Section

### 2.1   Chemicals

The following chemicals were used in this study: glyoxal solution (Sigma-Aldrich) 40 % in $H_2O$ (by weight), hydrogen peroxide (Kanto Chemical Co. Inc.) 30 % in $H_2O$ (by weight), ammonium sulfate (Sigma-Aldrich)

99.999% (by weight), sulfuric acid (Junsei Chemical Co. Inc.) 95.0 % in $H_2O$ (by volume), and nitric acid (Sigma Aldrich) 70% in $H_2O$ (by weight).

### 2.2   Environmental Chamber Method



Photooxidation/dark experiments for AS aerosols and SA aerosols were conducted in a 5 m$^3$ PTFE environmental chamber at Korea Institute Science and Technology (KIST), Seoul, Republic of Korea. The chamber was initially filled with clean/dry air (< 0.5 ppb NMHC, < 5 % RH) at 20-25ºC and atmospheric pressure. AS aerosols were made by atomizing the aqueous solution of 1 mM glyoxal, 200 μM $H_2O_2$, and 0.012 M $(NH_4)_2SO_4$ and SA aerosols were made by atomizing the aqueous solution of 1 mM glyoxal, 200 μM $H_2O_2$, and 0.012 M $H_2SO_4$. Note that atomized particles were passed through a diffusion dryer (3062-NC, TSI; residence time ~ 5 s) before introducing to the chamber to minimize the water content. 200 μM $H_2O_2$, previously used by Nguyen et al. (2013) in flow tube studies, was used here since this concentration generates ~$10^{-14}$ M, an atmospheric aqueous OH concentration (Arakaki et al., 2013), during the reaction with 1 mM glyoxal according to the updated full kinetic model (Lim et al., 2010; Lim et al., 2013; Lim and Turpin, 2015) (Fig. S1; Model details are in Section 3.3). The smog chamber background levels were < 40 particles and ~$10^{-2}$ μg/m$^3$. Atomized aerosols with mass concentrations of 70–180 μg/m$^3$ and a size of 70–90 nm (geometric mean diameter) were then introduced into the chamber for photooxidation and dark reactions.

The photooxidation was initiated by turning on blacklights, which generate tropospheric UV light (340—400 nm) with a peak intensity at 359 nm. For humid condition experiments, the KIST humidifier was used to achieve up to ~ 90 % RH. RH and temperature were measured using a Kimo AMI 300 multi function meter. All experiments were conducted in moderate $O_3$ and $NO_x$ conditions ($[O_3]_{initial}$ = 0–13 ppb, $[NO]_{initial}$ = 2–81 ppb and $[NO_2]_{initial}$ = 0–5 ppb). NO and $NO_x$ concentrations were measured using an NO-NO$_2$-NO$_x$ Analyzer (Thermo Scientific Model 42i). Ozone concentrations were measured using an Ozone Analyzer (Thermo Scientific Model 49i). We conducted 14 experiments in an irradiated/dark chamber adjusting relative humidity from 5 % to 90 %. Table S1 summarizes experimental conditions. Fractions of aerosol liquid water (ALW) in aerosols were estimated using the extended aerosol inorganic model (E-AIM) (Wexler and Clegg, 2002) and also included in the table. For the ALW estimations, Model II ($H^+$, $NH_4^+$, $SO_4^{2-}$, $NO_3^-$, $H_2O$, and Organic Compound) was used and neither radical nor non-radical reactions in the condensed phase were considered.

## 2.3 Filter Extraction and Aerosol Analysis

At chamber reaction time of 0, 30, 60, 120, and 180 minutes, aerosols were collected on blank Teflon filters (25 mm, 1.0 μm, Pall Corporation) for 10 minutes at a sampling rate of 10 LPM (Only 0 and 180 minute samples were collected for dark reactions). These filter samples were extracted with 5 mL Milipore water (18.2 MΩ) for 20 minute sonication. Ultra performance liquid chromatography coupled to quadrupole time-of-flight mass spectrometry (UPLC-Q-TOF-MS) (Waters, Synapt G2) was used to examine the elemental composition of aerosol samples. The capillary voltage and the capillary temperature were -2500 V and 350 °C, respectively for negative mode analyses. Positive mode analyses were conducted with the capillary voltage of 2500 V and the capillary temperature of 250 °C. The aerosol samples were diluted with methanol by 2 fold by volume (i.e., 50:50 water/methanol), then immediately introduced into the electrospray ionization source by direct infusion with a flow rate of 2.54 mL/hr. Since no column was used for separation, clusters could be formed during the ionization.

Two standard solutions, AS solution (1 mM glyoxal, 200 μM $H_2O_2$, and 0.012 M $(NH_4)_2SO_4$) and SA solutions (1 mM glyoxal, 200 μM $H_2O_2$, and 0.012 M $H_2SO_4$), were also analyzed by UPLC-Q-TOF-MS using the method described above.





Organic compounds (CHO), organonitrogens (CHNO), organosulfates (CHOS), and organosulfate-nitrogens (CHNOS) were analyzed both in the negative and positive modes. In the negative mode, deprotonated acid compounds (i.e., [M-H]$^-$) such as carboxylic acids, organonitrates and organosulfates, are detected. And some organonitrates are detected as Cl$^-$ adducts. In the positive mode, glyoxal, glyoxal oligomers and reduced

organonitrogen compounds (e.g., imines; note that imidazoles are imines) are detected via protonation ([M + H]$^+$) or as sodium adducts ([M + Na]$^+$). Elemental formulas were assigned by MIDAS Formula Calculator (version 1.2.3) within the uncertainty of 150 ppm based on the mass accuracy of measured nitrate peaks (m/z$^-$ 62). In addition to this, the elemental formulas for organosulfates (with 32S) were confirmed by the coexistence of the identical formula with the sulfur isotope (34S), present with a signal intensity reduced to ~ 100 times smaller in a

mass spectrum (Note that natural abundance is 95 % 32S and 4 % 34S). Similarly, Cl$^-$ adducts (35Cl) were confirmed by chlorine isotope (37Cl) adducts with an intensity reduced by ~ 20 times (Note that natural abundance is 75 % 35Cl and 25 % 37Cl).

### 2.4    Analysis for m/z$^-$ 147 and 226

The smog chamber product peak at m/z$^-$ 147 was further analyzed by a liquid chromatograph tandem mass

spectrometer (LC-MS/MS; 6460 Agilent Triple Quadrupole). Again, the mobile phase was 50:50 water/methanol and the direct injection with a flow rate of 0.1 mL/min (no column) was used. The sample was analyzed in the negative mode. The capillary voltage and the capillary temperature were -3000 V and 350 °C, respectively. A standard solution prepared by mixing glyoxal (7.6 mM) and nitric acid (15 mM) was analyzed by UPLC-Q-TOF-MS with the method described above.

## 3    Results and Discussion

### 3.1    Photochemical Organonitrate Formation in Wet Aerosols

During the irradiation for both AS and SA aerosols in the humid chamber (> 70 % RH) the major products, organonitrates (m/z$^-$ 147 and 226) were likely formed by non-radical reactions of glyoxal with nitric acid in the aqueous phase of wet aerosols (Fig.1A and B), and nitric acid (m/z$^-$ 62) is formed via radical reactions in the gas-

phase system of glyoxal-NO$_x$-OH. LC-MS/MS analysis was conducted for m/z$^-$ 147, and fragments were m/z$^-$ 62 and 103 (Fig. S2). m/z$^-$ 62 indicates nitrate acid and m/z$^-$ 103 is a loss of 44 (CO$_2$) suggesting that the parent molecule is a carboxylic acid. The standard solution of the glyoxal-HNO$_3$ mixture analyzed by UPLC-Q-TOF-MS showed the major peaks at m/z$^-$ 62, 147 and 226 (Fig. S3). This confirms that m/z$^-$ 147 and 226 can be formed via aqueous non-radical reactions of glyoxal and nitric acid without UV. According to the MIDAS molecular

calculator, these two peaks are organonitrates (m/z$^-$ 147, C6H2N2O12 (z = 2); m/z$^-$ 226, C4H1N1O8Cl1), not likely nitric acid adducts to glyoxal (clusters). The proposed formation and molecular structures are illustrated in Scheme S1. It appears that nitric acid undergoes nitrate ester formation (R-OH + HNO$_3$ → R-ONO$_2$ + H$_2$O) (Boschan et al., 1955) and oxidizes some hydroxyl groups (Connelly et al., 2012). Therefore, m/z$^-$ 147 and 226 are organonitrates formed by the aqueous-phase reaction of glyoxal and nitric acid. m/z$^-$ 147 is likely to be doubly

charged (z = 2) and this is supported by the coexistence of m/z$^-$ 147.5 (Δz = 0.5). On the other hand, m/z$^-$ 226 is likely to be a Cl$^-$ adduct organonitrate. Cl$^-$ adducts for organonitrates have been observed (Bouma and Jennings, 1981; Lawrence et al., 2001; Rajapakse et al., 2016; Zhu and Cole, 2000) while MIDAS does not propose realistic





organonitrates without Cl⁻. The uncertainty for the mass of Cl adducts is reasonably low (~ 50 ppm). Cl⁻ adducts are confirmed by the coexistence of m/z⁻ 228 ($\Delta z = 2$), organonitrates adducted by Cl⁻ isotope (37Cl) with the low mass uncertainty (10 – 30 ppm).

Despite the non-radical formation of organonitrates, nitric acid was photochemically formed during the irradiation
(Scheme 1A). It is also possible that nitric acid is formed in the dark (Scheme 1B). Indeed, m/z⁻ 62, 147 and 226 initially appeared for AS and SA aerosols in dark reactions (Fig. 2E and F). However, these peaks disappeared in 3 hours (Fig. S4). This suggests that photochemical formation of nitric acid is continuous during the irradiation. The chamber was initially filled with NO and little $NO_2$, but NO was converted to $NO_2$ as irradiation proceeded. Peroxy radicals are effective for the $NO$-$NO_2$ conversion (Atkinson and Arey, 2003) and our measured $NO_x$ levels
support this. During the irradiation of AS aerosols (Experiment #1, Table S1), the concentration of $NO_2$ increased from 4.0 ppb to 20.0 ppb ($\Delta[NO_2] = 16.0$ ppb) while the concentration of NO was reduced from 24.0 ppb to 5.5 ppb ($\Delta[NO] = -18.5$ ppb). Another photochemical experiment for AS aerosols (Experiment #2) also shows the significant increase of $[NO_2]$ ($\Delta[NO_2] = 10.0$ ppb, $\Delta[NO] = -3.8$ ppb). However, in the photochemical experiment for AS aerosols containing no glyoxal (Experiment #7), $NO_2$ only increased by 0.7 ppb while the initial $[NO]$ was
similar to Experiment #2. Notice that $O_3$ increased. This is due to photolysis of $NO_2$.

Then, which organic species became peroxy radicals in the gas phase? Glyoxal is not likely to evaporate due to high water solubility (effective $H_{glyoxal} \sim 2e7$) (Ervens and Volkamer, 2010). Among products of glyoxal-OH reactions glyoxylic acid and formic acid are not very water soluble ($H_{glyoxylic\ acid} = 9.12e3$ M/atm; $H_{formic\ acid} = 5.50e3$), so they could evaporate to the gas phase and undergo OH radical reactions forming peroxy radicals
(Model simulations are discussed in Section 3.3). OH radicals produced via photolysis of $H_2O_2$ could evaporate while reacting with glyoxal in the aqueous phase ($H_{OH} = 30$ M/atm). Scheme 2 shows gas-phase OH reactions of glyoxylic acid and formic acid. In the gas phase, glyoxylic acid (HO(O)CC(O)H) reacts with OH and $O_2$ forming peroxy radicals (HO(O)CC(O)OO·), which convert NO to $NO_2$ (HO(O)CC(O)OO· + NO → HO(O)CC(O)O· + $NO_2$). Although there is no literature available for OH reactions of glyoxylic acid in the gas phase, these peroxy
radicals (HO(O)CC(O)OO·) are not likely to produce organic nitrates since the similar structured peroxy radicals (H(O)CC(O)OO·), which are formed from OH reactions of glyoxal in the gas phase, are reported to produce neither alkyl nitrates (H(O)CC(O)OO· + NO → H(O)CC(O)ONO₂) nor alkyl peroxyacetylnitrates (H(O)CC(O)OO· + NO₂ → H(O)CC(O)OONO₂), and only convert NO to $NO_2$ (Orlando and Tyndall, 2001). OH radical reactions of formic acid lead to $CO_2$ without $NO_x$ reactions (Scheme 2B) (Atkinson et al., 2004). Therefore,
glyoxylic acid is likely to be the source of peroxy radicals that convert NO to $NO_2$, and organonitrates in this work are not from gas-phase formation. Since gas-phase OH reactions of glyoxylic acid and formic acid produce $HO_2$ (Scheme 2A and B), this $HO_2$ contributes to additional conversion from NO to $NO_2$ and recycles OH (i.e., $HO_2$ + NO → OH + $NO_2$) (Orlando and Tyndall, 2001). Lastly, $NO_2$ reacts with OH forming $HNO_3$ (Scheme 1A) (Finlayson-Pitts and Pitts Jr, 1999).

Subsequently, nitric acid is taken up into ALW. In the humid chamber, estimated ALW fractions in wet aerosols were 19.4—45.8 % for AS aerosols (Experiment #4, Table S1) and 53.1—74.2 % for SA aerosols (Experiment #12) throughout experiments, and after the 3 hour irradiation the peaks at m/z⁻ 62 (nitric acid), 147 and 226 were prominent (Fig. 1A and B). However, in the dry chamber, ALW fractions were only ~ 1 % for AS aerosols





(Experiment #1) and 26.5—32.4 % (Experiment #11) for SA aerosols. After the 3 hour irradiation, only m/z⁻ 62 and 147 (m/z⁻ 226 was missing) appeared with smaller intensities for AS aerosols (Fig. 1C), and none of m/z⁻ 62, 147, and 226 appeared for SA aerosols (Fig. 1D). It is interesting that $HNO_3$ was found in AS aerosols in the dry chamber. Formation of $HNO_3$ by heterogeneous reactions of $NO_2$ on the surface of aerosols has been reported previously (Crowley et al., 2010). Note, the oxidation of NO (and $NO_2$) by $O_2$ in the gas phase is too slow (Atkinson et al., 2004), and NO and $NO_2$ are not very soluble for partitioning into the aqueous phase ($H_{NO}$ = 0.019 M/atm, $H_{NO2}$ = 0.007 M/atm). Therefore, heterogeneous reactions $NO_2$ on the dry surface of AS aerosols could form $HNO_3$. No nitric acid was observed in SA aerosols, which still contain 26.5—32.4 % ALW in the dry chamber. It is possible that in the presence of sulfuric acid nitric acid acts as a base forming $NO_2^+$ and $HSO_4^-$ (i.e., $HNO_3 + 2H_2SO_4 \rightarrow NO_2^+ + 2HSO_4^- + H_3O^+$) (McQuarrie et al., 1991). Further studies are required for $HNO_3$ uptake by AS and SA aerosols in the dry chamber.

Figure 2 also suggests gas phase photochemical production and uptake of $HNO_3$ into ALW. In Fig. 2A and B, both AS and SA solutions only show sulfuric acid peaks at m/z⁻ 97 (monomer) and m/z⁻ 195 (dimer), and an organosulfate peak at m/z⁻ 217 (C2H1O8S2), which is an ester product of a glyoxal and two sulfuric acids with the structure (A) and the formation (B) in Scheme 3. In the dry and dark chamber, neither nitric acid (m/z⁻ 62) nor organonitrates (m/z⁻ 147 and 226) were initially formed (Fig. 2C and D) suggesting little $HNO_3$ uptake in the dry chamber. Note that in the dry chamber ALW fractions are 1 % for AS aerosols (Fig. 2C; Experiment #8, Table S1) and 27 % for SA aerosols (Fig. 2D; Experiment #13). Again, in the humid and dark chamber m/z⁻ 62, 147 and 226 initially appeared for AS and SA aerosols (Fig. 2E and F) and this is due to $HNO_3$ uptake into sufficient ALW in both aerosols (ALW fraction for AS and SA aerosols are 54 %, 71 % respectively). But $HNO_3$ here is formed by dark reactions of $O_3$ and $NO_x$ (Scheme 1B) and disappeared in the 3 hour dark reactions (Fig. S4).

### 3.2 Dilute Cloud-Relevant (Bulk) Chemistry vs. Multiphase Aerosol Photooxidation

The photochemistry of glyoxal in dilute aqueous solution vessel has been established (Lim et al., 2010; Tan et al., 2010). The OH reaction of glyoxal in the aqueous phase produces mostly oxalic acid with minor products like glyoxylic acid, formic acid and carbonic acid. When the glyoxal concentration is higher than cloud-relevant concentration, tartaric acid also becomes a major product formed via organic radical-radical reactions. The reaction mechanisms including concentration-dependent pathways were elucidated, and the aqueous photochemical kinetic model (Lim et al., 2010; Lim et al., 2013) was developed and validated by experimental results. Assuming adequate access to OH radicals, the model (Perri et al., 2010) predicts that oxalic acid, tartaric acid, and organosulfates will form via radical reactions in $H_2SO_4$-containing wet aerosols. However, multiphase modeling suggests depletion of OH radicals in wet aerosols may be substantial and predicts that non-radical chemical pathways will dominate, leading to the formation of organosulfates (McNeill et al., 2012).

While sulfuric acid and ammonium hydroxide do not interfere the real-time formation of oxalic acid in dilute (cloud-relevant) photooxidation experiments (bulk) (Ortiz-Montalvo et al., 2014; Tan et al., 2009), non-radical reactions with sulfate and ammonium ions dominate in AS and SA aerosols during the 3 hour irradiation in the humid/dry chamber. Mass spectral analyses in the negative mode (Fig. 1) suggest that the products are organic acids (CHO), organonitrates (CHNO), organosulfates (CHOS), and nitrooxy-organosulfates (CHNOS). Proposed chemical formula are listed in Table 1.





However, production of oxalic acid and tartaric acid provides evidence that OH reactions of glyoxal occurred in ALW. Note that oxalic acid cannot be formed in the gas phase since gas-phase photochemistry will decompose evaporated organic species (e.g., glyoxylic acid, formic acid) to $CO_2$ (Scheme 2). During the irradiation, oxalic acids were formed in the humid chamber since UPLC-Q-TOF-MS detected m/z⁻ 89 (oxalic acid) in AS aerosols

(Fig. 1A) and m/z⁻ 187 (oxalic acid-sulfuric acid adduct) in SA aerosols (Fig. 1B). Sulfuric acid adducts to organic acid are commonly observed during aerosol nucleation and particle growth (Zhang et al., 2004). Even in the dry chamber oxalic acid (m/z⁻ 89), glycolic acid (m/z⁻ 173 as a sulfuric acid adduct) and tartaric acid (m/z⁻ 247 as a sulfuric acid adduct) were formed as in SA aerosols (Fig. 1D). This can be explained by the high hygroscopicity of sulfuric acid. In the dry chamber (7 % RH) SA aerosols still held 32 % ALW (Experiment #11, Table S1).

The decay of glyoxal provides additional evidence that glyoxal reacts with OH radicals in ALW. Glyoxal is detected in the positive mode of UPLC-Q-TOF-MS. A number of peaks at m/z⁺ 59 ($= [M + H]^+$), 95 ($= [M + 2H_2O + H]^+$), 99 ($= [M + H_2O + Na]^+$), 113 ($= [M + MeOH + Na]^+$), 117 ($= [M + 2H_2O + Na]^+$), and 145 ($= [M + 2MeOH + Na]^+$) represent glyoxal in various hydrated forms and hemiacetal forms with water and methanol from the mobile phase (M = glyoxal $(CHO)_2$; MeOH = methanol). In Fig. 3, glyoxal peaks in AS aerosols and SA

aerosols are plotted in a relative scale (no glyoxal peak was found for AS aerosols in the humid chamber). The relative intensity was obtained by normalizing the fraction of the raw signal intensity divided by the weight of the collected particles on the filter. These estimations are not based on real-time online analyses since extracted filters were collected for 10 minutes, but still qualitatively indicate the glyoxal decay in wet aerosols. For SA aerosols, the decay rate of glyoxal in the dry chamber (0.03 min⁻¹ in Fig. 3A) is similar to that in the humid chamber (0.02

min⁻¹ in Fig. 3B) due to high hygroscopicity of sulfuric acid (32% ALW in the dry chamber). Assuming no evaporation of ALW, the kinetic model (Details are discussed in the next section) predicts that the decay rate of glyoxal by OH reactions in the aqueous phase is 0.018 min⁻¹, which is very similar to estimated values above. However, for AS aerosols in the dry chamber, glyoxal peaks at m/z⁺ 113, 117 and 131 decay sharply in 30 minutes and the estimated decay rate is ∼ 0.09 min⁻¹ (Fig. 3C), which is ∼ 5 times faster than the decay rate by OH reactions.

Since AS aerosols in the dry chamber hold only ∼ 1 % ALW, this suggests that ALW evaporation affects glyoxal loss significantly. So it is possible that gas-phase glyoxal chemistry takes place during the irradiation of AS aerosols in the dry chamber since glyoxal could evaporate, too. However, gas-phase photochemistry of glyoxal produce neither oxalic acid nor organonitrates (PAN type compounds); it produces decomposed fragments (Scheme 2C) (Atkinson et al., 2006; Orlando and Tyndall, 2001).

Organonitrogens (CHNO), organosulfurs (CHOS), organonitrogen-sulfur (CHNOS) and organic compounds (CHO) were also detected in the positive mode of UPLC-Q-TOF-MS (Table 2 and Fig. 4). Imidazoles (m/z⁺ 69, 145, 149, and 203) observed by Kampf et al. were also observed here in AS aerosols in the humid and dry chamber.

Organosulfates were formed in both AS and SA aerosols and detected in the negative mode of UPLC-Q-TOF-MS. The organosulfate (m/z⁻ 155, C2H3O6S1) observed by Galloway et al. was also observed in AS aerosols in

the humid chamber. Galloway et al. proposed two structures for m/z⁻ 155, a glyoxal-sulfate and a glycolic acid-sulfate ester and we argue that it is more likely to be the glycolic acid-sulfate ester. Since OH reactions of glyoxal produce glycolic acid in the presence of $HO_2$, which is commonly available during aqueous photochemistry (Lim and Turpin, 2015) (Scheme 4A), the glycolic-acid sulfate ester is formed by non-radical esterification between glycolic acid and sulfuric acid (Scheme 4B). The m/z⁻ 173 organosulfate (C2H5O7S1) formed in SA aerosols in




the dry chamber (7 % RH, 32 % ALW) is likely to be a glycolic acid-sulfuric acid adduct (Scheme 4C). However, the m/z⁻ 171 (organosulfate, C2H3O7S1), which was formed via glycolic acid radical-sulfuric acid radical reactions (Perri et al., 2010), was not observed here. This suggests that in the condensed phase OH radicals mostly contribute to oxidation of organic compounds, making products that subsequently form organosulfates or organic

acid-sulfuric acid adducts via non-radical reactions.

### 3.3     Model Simulations for Smog Chamber Photooxidation

In this work, aqueous glyoxal chemistry described in a previous kinetic model (Lim et al., 2010; Lim et al., 2013; Lim and Turpin, 2015) was expanded by including partitioning of radical oxidants (e.g., OH) and organic compounds (e.g., glyoxylic acid, formic acid) into the gas phase and the gas-phase OH reactions. Newly added

reactions are listed in Table S3. The model was then applied to better understand the multiphase chemistry in the smog chamber experiments, including whether the proposed pathway for the production of $NO_2$ and subsequent organonitrate formation is plausible. It is assumed that carboxylates (e.g., formate, glyoxylate, and oxalate) do not evaporate since they are likely to form low volatility carboxylate salts in the atmosphere (Ortiz-Montalvo et al., 2012), so only deprotonated acids evaporate according to water solubility. The model is not well designed to

simulate the dry conditions for AS aerosols in the chamber; since water evaporation is not allowed.

With the same initial concentrations of glyoxal (1 mM), $H_2O_2$ (200 μM) and $(NH_4)_2SO_4$ or $H_2SO_4$ (0.012 M), the model predicts that the dominant product is glyoxylic acid in both AS and SA aerosols in the presence of ALW (Fig. 5). In dilute (bulk) aqueous chemistry experiments designed to study chemistry in cloud water, the dominant product was found to be oxalic acid (Tan et al., 2009). But in the chamber, where aqueous chemistry takes place

in concentrated non-ideal solutions in wet aerosols with large surface to volume ratios, oxalic acid formation was suppressed by partitioning of glyoxylic acid to the gas phase, consistent with predictions elsewhere (Skog et al., submitted). The concentration of gas-phase glyoxylic acid produced through AS aerosol-phase chemistry reached up to ~ 200 ppb from AS aerosols (Fig. 6A) and ~ 800 ppb from SA aerosol-phase chemistry (Fig. 6B) in the presence of ALW. Note the predicted OH concentration in the gas phase was ~1e6 molecules/cm³, which is

atmospherically-relevant (Fig. S5). Clearly, in the gas phase sufficient amounts of peroxy radicals would have been formed from aqueous aerosol photochemistry to convert NO to $NO_2$.

### 3.4     Dark Aerosol-Phase Reactions

AS and SA solutions, and AS and SA aerosols at t = 0 min in the dark chamber were analyzed by UPLC-Q-TOF-MS in the negative mode (Fig. 2). Solutions only show sulfuric acid and organosulfate (Fig. 2A and B), but

atomized aerosols show many other peaks including organic acids (CHO), organosulfates (CHOS), nitric acids (HNO₃), organonitrogens (CHNO) and organonitrogen-sulfates (CHNOS) (Fig. 2C, D, E and F). When the solutions are atomized and introduced into the smog chamber, water evaporates from aerosols due to the large increase of the surface area, and concentrations of solutes increase. Water loss and concentration increase result in the formation of oligomers and inorganic products. Glyoxal forms oligomers via hemiacetal formation and

aldol-condensation can lead organic acid products (Lim et al., 2010; Loeffler et al., 2006; Sareen et al., 2010). Generally, organic acid oligomers (CHO) were formed in the dry chamber, and inorganic compounds and oligomers (CHNO, CHNOS) in the humid chamber (Proposed molecular formula are listed in Table 3). It appears that acid catalyzation (aldol condensation and hemiacetal formation) leading to organic acid formation favors low





ALW and high acidity because a hydrated glyoxal in the aqueous phase will be partially dehydrated to form an aldehyde group, and the dehydration can be maximized by the evaporation of ALW in the dry chamber (Lim et al., 2010). However, acidity effects on oligomer formation requires further study. As discussed in Section 3.1, the prominent peaks represent nitric acid (m/z⁻ 62) and organonitrates (m/z⁻ 147 and 226) for both AS and SA aerosols

in the humid dark chamber (Fig. 2E and F). In the dark, nitric acid is formed by $N_2O_5$ uptake by water and the gas-phase reaction of $NO_2$ and $NO_3$ produces $N_2O_5$ (Scheme 1B) (Finlayson-Pitts and Pitts Jr, 1999). Dark experiments were conducted initially with NO and $O_3$ available, so $NO_2$ and $NO_3$ were probably formed by the $O_3$ oxidation of NO and $NO_2$, respectively (Scheme 1B).

After 1 hour of dark reactions, m/z⁻ 62, 147 and 226 disappeared in AS and SA aerosols in the humid chamber

(Fig. S4). $NO_x$ and $O_3$ levels stayed almost the same indicating little $HNO_3$ formation in the dark. Instead, many other m/z⁻ peaks appeared indicating that various non-radical reactions took place. Non-radical reactions also occurred in the dry chamber.

In the positive-mode mass spectra for the AS solution, imidazole (m/z⁺ 69), glyoxal (m/z⁺ 117), and imines (m/z⁺ 133, 248, and 363) were detected (Fig. S6A) while high molecular weight organic compounds, which are

presumably acid-catalyzed products from glyoxal, were detected in the SA solution (Fig. S6B). Many more peaks are found in the mass spectra of AS and SA aerosols (Fig. S6C, D, E and F) than the solutions. Proposed elemental compositions based on MIDAS are listed in Table S2. All organonitrogens in the positive mode for AS aerosols are expected to be imines because in the aqueous phase reactions of glyoxal with ammonium form imines (Galloway et al., 2009; Noziere et al., 2009; Yu et al., 2011). Organic compounds (CHO) are oligomers of glyoxal,

which are mostly detected in SA aerosols while imines (CHNO) are prominent in AS aerosols.

After 1 hour dark reactions, oligomerization was evident in the positive mode mass spectra from dark experiments (Fig. S7). It appears that in AS aerosols ammonium ions form oligomers and imines, whereas in SA aerosols the formation of oligomers and organosulfates are enhanced by sulfuric acid. This is consistent with the previous observation of oligomer formation for IEPOX (Nguyen et al., 2014). When ALW is sufficient (in AS aerosols in

the humid chamber and SA aerosols in the humid/dry chamber), mass spectra are similar (Fig. S7A, B, and D), in which unidentified peaks at m/z⁺ 99, 261, 299, 301, and 305 are dominant, indicating products of multiphase aerosol reactions are different from those of heterogeneous reactions on the dry aerosol surface (AS aerosols in the dry chamber, Fig. S7C). Further studies are required for surface and multiphase chemistry affected by hygroscopicity and acidity of aerosols and RH.

**4    Conclusions and Atmospheric Implications**

This work demonstrates $HNO_3$ uptake by wet aerosols and formation of organonitrates with water-soluble organic constituents via aqueous chemistry, which to our knowledge has not been reported previously. Organonitrates formed in aerosol waters are not likely alkylnitrates formed by peroxy radical reactions with NO in the gas phase, followed by gas-particle partitioning (Lee et al., 2016) because yields of alkylnitrates in alkane-OH-$NO_x$ chamber

experiments decreased in humid conditions due to hydrophobicity (Lim and Ziemann, 2009). In order to be hydrophilic, VOCs must contain a small number of carbons. This small size also facilitates uptake into wet aerosols. The reaction of peroxy radicals with NO favors alkoxy radical formation, and suppresses alkylnitrate formation (Arey et al., 2001). In our irradiated chamber, this organonitrate chemistry was facilitated by gas phase





VOC-NO$_x$-O$_3$ photochemistry (Table S1) forming HNO$_3$. This represents typical anthropogenic photochemistry of VOC-NO$_x$-O$_3$ leading to HNO$_3$ formation as a sink (Finlayson-Pitts and Pitts Jr, 1999). Note concentrations of glyoxal and hydrogen peroxide in atomized solutions are atmospherically-relevant (Guo et al., 2014; Lim et al., 2013).

Nitrates are major constituents in atmospheric aerosols (Zhang et al., 2007). They are very hygroscopic, facilitating water uptake into aerosols (Hennigan et al., 2008a; Hodas et al., 2014). ALW in turn facilitates aerosol partitioning of HNO$_3$. Field observations support HNO$_3$ uptake by aerosol waters during the daytime (Allen et al., 2015; Hodas et al., 2014). And notably, nitrate concentrations are strongly correlated with water soluble organic compound concentrations (Hennigan et al., 2008b). This study suggests that small, highly oxidized organonitrates

are formed in wet aerosols. An understanding of their contribution to overall atmospheric organonitrate particulate mass warrants further study.

## 5   Acknowledgements

This work was supported by Brainpool Fellowship from Ministry of Science, ICT and Future Planning, Republic of Korea (152S-5-2-1416) and Korea Institute of Science and Technology. The authors acknowledge Minsuk Oh

and Hyunmi Park at Mass Spectrometry Division in KIST Chemical Analysis Center for UPLC-HR-Q-TOF-MS analysis. The authors thank Jino Kim for laboratory assistance. The authors also thank Frank N. Keutsch for helpful discussions. Barbara J. Turpin gratefully acknowledges support from the US National Science Foundation (#10526611).

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



**Table 1: Elemental compositions of organic-inorganic compounds at 180 minute irradiation time (UPLC-Q-TOF-MS negative mode analysis)**

| Aerosols (Conditions) | m/z⁻ | Elemental Composition | Compound | Error (ppm) |
|---|---|---|---|---|
| AS (Humid) | 61.9907 | N1O3 | Nitric Acid | 37.6 |
| | 88.9840 | C2H1O4 | Oxalic Acid | 45.3 |
| | 96.9667 | H1O4S1 | Sulfuric Acid | 68.0 |
| | 146.9669 (z = 2) | C6H2N2O12 | Organonitrate | -95.5 |
| | 154.9581 | C2H3O6S1 | Glycolic acid-sulfate Ester* | -48.3 |
| | 197.9096 | C2N1O8S1 | Nitrooxy-organosulfate | -128.4 |
| | 225.9279 | C4H1N1O8Cl(35)1 | Organonitrate | -51.9 |
| | 227.9426 | C4H1N1O8Cl(37)1 | | 26.0 |
| SA (Humid) | 61.9911 | N1O3 | Nitric Acid | 44.1 |
| | 96.9603 | H1O4S1 | Sulfuric Acid | 2.0 |
| | 146.9665 (z = 2) | C6H2N2O12 | Organonitrate | -98.2 |
| | 181.9389 | C2N1O7S1 | Nitrooxy-organosulfate | -6.6 |
| | 186.9598 | C2H3O8S1 | Oxalic Acid-Sulfuric Acid | 23.5 |
| | 197.9053 | C2N1O8S1 | Nitrooxy-organosulfate | -150.1 |
| | 225.9278 | C4H1N1O8Cl(35)1 | Organonitrate | -52.3 |
| | 227.9404 | C4H1N1O8Cl(37)1 | | 16.4 |
| | 282.8897 | C6H3O11S1 | Organosulfate | -178.4 |
| | 288.8996 | C8H1O10S1 | Organosulfate | -103.8 |
| AS (Dry) | 61.9925 | N1O3 | Nitric Acid | 66.7 |
| | 96.9614 | H1O4S1 | Sulfuric Acid | 13.4 |
| | 146.9677 (z = 2) | C6H2N2O12 | Organonitrate | -90.1 |
| | 171.0956 | C6H11N4O2 | Organonitrogen | 40.0 |
| | 173.0084 | C6H5O6 | Organic Acid | -4.4 |
| SA (Dry) | 62.0241 | N1O3 | Nitric Acid | -10.5 |
| | 89.0389 | C2H5N2O2 | Organonitrogens | 36.5 |
| | 96.9604 | H1O4S1 | Sulfuric Acid | 3.1 |
| | 172.9572 | C2H5O7S1 | Glycolic Acid-Sulfuric Acid | 109.5 |
| | 186.9687 | C2H3O8S1 | Oxalic Acid-Sulfuric Acid | 71.1 |
| | 247.0045 | C4H7O10S1 | Tartaric Acid-Sulfuric Acid | 113.2 |

*Glycolic acid-sulfate ester was detected by Galloway et al., 2009



**Table 2:** **Elemental compositions of glyoxal and other organic-inorganic compounds at 180 minute irradiation time (UPLC-Q-TOF-MS positive mode analysis)**

| Aerosols (Conditions) | m/z$^+$ | Elemental Composition | Compound | Error (ppm) |
|---|---|---|---|---|
| AS (Humid) | 69.0491 | C3H5N2 | Imidazole* | 63.4 |
| | 107.9732 (z = -2) | C4H1O9Na1 | Organic Peroxide | -20.0 |
| | 109.0698 | C4H10N2Na1 | Imine | -35.0 |
| | 145.0652 | C5H9N2O3 | Imidazole* | 30.5 |
| | 149.0299 | C5H6N2O2Na1 | Imidazole | -72.8 |
| | 203.1056 | C7H12N6Na1 | Imidazole | 19.9 |
| SA (Humid) | 95.0307 | C2H7O4 | Glyoxal** (dihydrated) | 33.5 |
| | 98.9911 | C2H4O3Na1 | Glyoxal** (monohydrated) | -143.1 |
| | 145.0499 | C4H10O4Na1 | Glyoxal (hydrated by 2 MeOHs) | 19.1 |
| | 149.0294 | C6H6O3Na1 | Organic Compound | 56.9 |
| AS (Dry) | 69.0528 | C3H5N2 | Imidazole* | 116.9 |
| | 113.0274 | C3H6O3Na1 | Glyoxal (hydrated by 1 MeOH) | 57.4 |
| | 117.0098 | C2H4O6Na1 | Glyoxal (dihydrated) | -51.5 |
| | 131.0348 | C3H8O4Na1 | Glyoxal (hydrated by 1 MeOH) | 25.3 |
| | 149.0299 | C5H6N2O2Na1 | Imidazole | -33.9 |
| | 203.0614 | C7H11N2O5 | Imidazole* | 23.9 |
| SA (Dry) | 95.0415 | C2H7O4 | Glyoxal** (dihydrated) | 80.1 |
| | 98.9894 | C2H4O3Na1 | Glyoxal** (monohydrated) | -160.3 |
| | 131.0065 | C3H8O4Na1 | Glyoxal (hydrated by 1 MeOH) | -190.7 |
| | 149.0213 | C6H6O3Na1 | Organic Compound | 2.6 |

*Imidazole detected by Kampf et al., 2012

**Glyoxal appeared at t = 0 min, but disappeared during the irradiation



**Table 3: Elemental compositions of organic-inorganic compounds in dark reactions (UPLC-HR-Q-TOF-MS negative mode analysis)**

| Aerosols (Conditions) | m/z⁻ | Elemental Composition | Compound | Error (ppm) |
|---|---|---|---|---|
| AS (Solution) | 96.9596 | H1O4S1 | Sulfuric Acid | -5.2 |
| | 194.9268 | H3O8S2 | Sulfuric Acid Dimer | -3.5 |
| | 216.9095 | C2H1O8S2 | Organosulfate | -10.8 |
| AS (Dry) | 96.9608 | H1O4S1 | Sulfuric Acid | 7.2 |
| | 216.9142 | C2H1O8S2 | Organosulfate | 10.9 |
| | 275.1671 | C13H23O6 | Organic Acid Oligomer | 62.1 |
| | 311.1689 | C13H27O8 | Organic Acid Oligomer | -7.2 |
| | 339.1974 | C15H31O8 | Organic Acid Oligomer | -14.9 |
| | 397.0972 | C14H21O13 | Organic Acid Oligomer | -3.9 |
| AS (Humid) | 61.9862 | N1O3 | Nitric Acid | -35.0 |
| | 96.9603 | H1O4S1 | Sulfuric Acid | 2.0 |
| | 146.9671 (z = 2) | C6H2N2O12 | Organonitrate | -94.1 |
| | 181.9377 | C2N1O7S1 | Nitrooxy-Organosulfate | -13.2 |
| | 197.9200 | C2N1O8S1 | Nitrooxy-Organosulfate | -75.8 |
| | 209.9507 | C3N1O8S1 | Nitrooxy-Organosulfate | 14.8 |
| | 225.9276 | C4H1N1O8Cl1 | Organonitrate | -52.7 |
| | 243.9025 | C3O9S2 | Organosulfate | 14.7 |
| | 288.9074 | C4H1O13S1 | Organosulfate | 24.0 |
| | 373.8744 | C6N1O14S2 | Nitrooxy-Organosulfate | -5.8 |
| | 401.9027 | C6N3O16S1 | Nitrooxy-Organosulfate | 5.5 |
| | 486.8823 | C10H3N2O17S2 | Nitrooxy-Organosulfate | -11.1 |
| SA (Solution) | 96.9614 | H1O4S1 | Sulfuric Acid | 13.4 |
| | 194.9283 | H3O8S2 | Sulfuric Acid Dimer | 4.2 |
| | 216.9104 | C2H1O8S2 | Organosulfate | -6.6 |
| SA (Dry) | 96.9611 | H1O4S1 | Sulfuric Acid | 10.3 |
| | 275.1615 | C13H23O6 | Organic Acid Oligomer | 41.7 |
| | 293.1608 | C13H25O7 | Organic Acid Oligomer | 0.8 |
| | 311.1589 | C13H27O8 | Organic Acid Oligomer | -39.3 |
| | 339.1808 | C13H27O9 | Organic Acid Oligomer | -43.5 |
| SA (Humid) | 61.9908 | N1O3 | Nitric Acid | 39.3 |
| | 96.9622 | H1O4S1 | Sulfuric Acid | 21.6 |
| | 146.9669 (z = 2) | C6H2N2O12 | Organonitrate | -95.5 |
| | 209.9499 | C10N2O15S1 | Nirooxy-Organosulfate | -7.7 |
| | 225.9277 | C4H1N1O8Cl1 | Organonitrate | -52.7 |
| | 243.9090 | C6O18S4 | Organonitrate | 41.3 |
| | 288.9016 | C4H1O11S2 | Organosulfate | 17.4 |
| | 311.1672 | C10H23N4O7 | Organonitrate | 32.1 |
| | 339.1943 | C12H27N4O7 | Organonitrate | 17.0 |
| | 373.8829 | C6N1O14S2 | Nitrooxy-organosulfate | 16.9 |
| | 401.9040 | C6N3O16S1 | Nitrooxy-organosulfate | 8.8 |
| | 486.8861 | C8H7O18S3 | Nitrooxy-organosulfate | 12.5 |





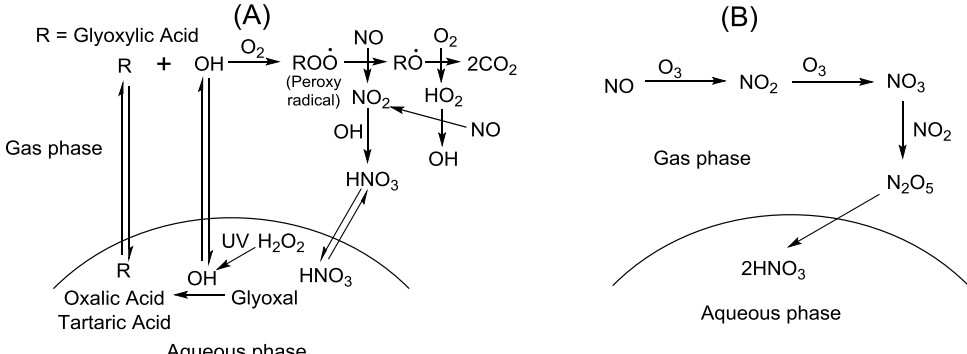

**Scheme 1: Nitric acid formation in the UV (A) and in the dark (B)**





**Scheme 2: Mechanisms of gas-phase OH reactions in the presence of NO$_x$ for glyoxylic acid (A) formic acid (B) and glyoxal (C)**





(A)

C2H1O8S2

(B)

Monohydrated Glyoxal

Monohydrated Glyoxal (Enol)

$H_2SO_4$

$H_2O$

$H_2SO_4$

$H_2O$

**Scheme 3:  The structure of C2H1O8S2 (A) and its formation from glyoxal and sulfuric acids (B)**




**Scheme 4: Proposed glycolic acid formation from OH reaction of glyoxal in the aqueous phase (A), glycolic acid-sulfate ester formation from non-radical reactions of glycolic acid with sulfuric acid (B), and glycolic acid-sulfuric acid adduct formation (C)**



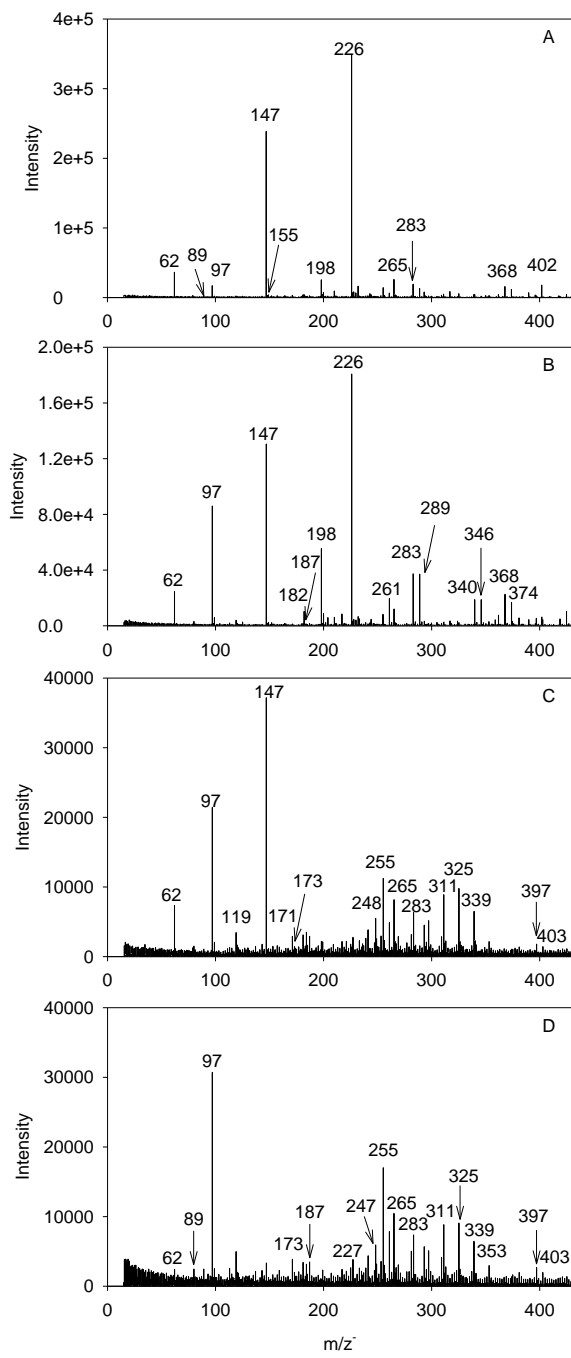

**Figure 1: Negative-mode mass spectra for 3 hour-irradiated AS aerosols (A) and SA aerosols (B) in the humid chamber, and AS aerosols (C) and SA aerosols (D) in the dry chamber**





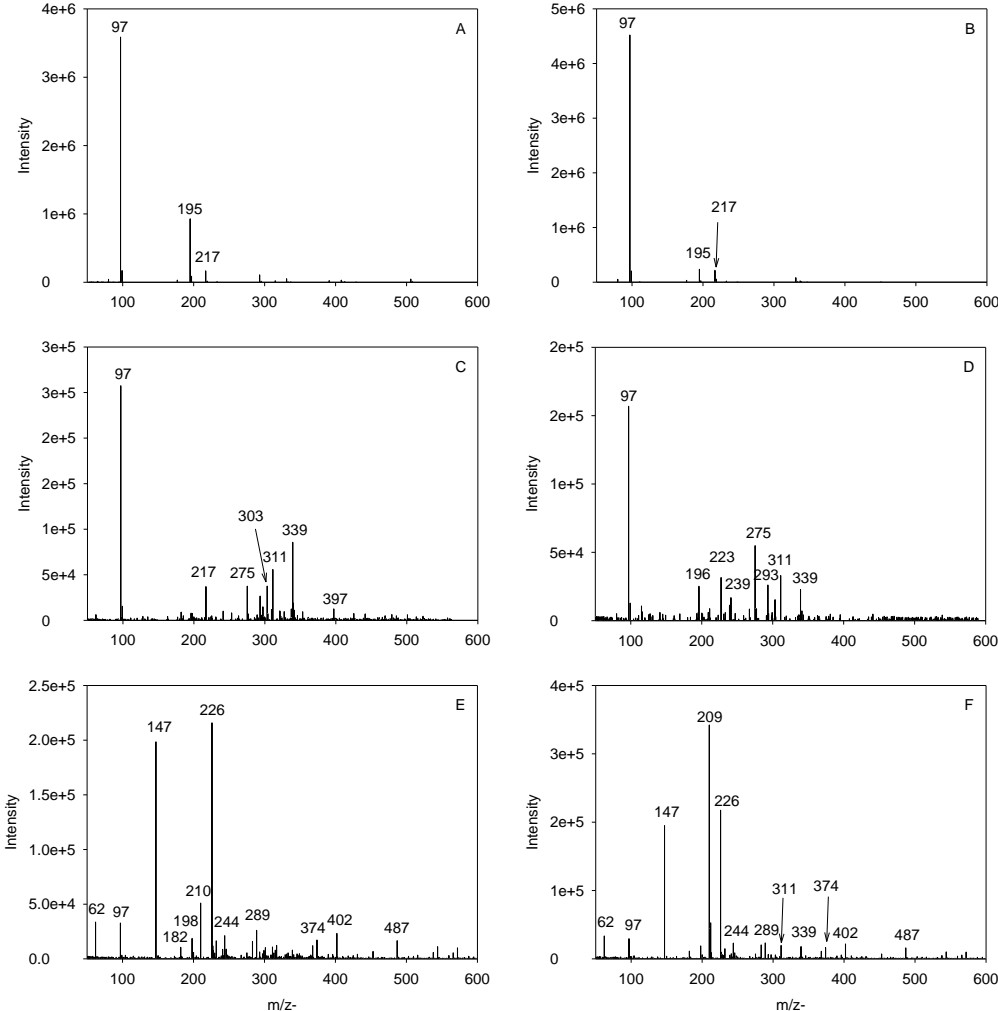

**Figure 2:** **Negative-mode mass spectra for dark reactions of glyoxal-$H_2O_2$-$(NH_4)_2SO_4$ (A, C and E) and glyoxal-$H_2O_2$-$H_2SO_4$ (B, D and F). (A) and (B) are solutions. (C) and (D) are aerosols in the dry chamber, and (E) and (F) are aerosols in the humid chamber at dark reaction time = 0 min**





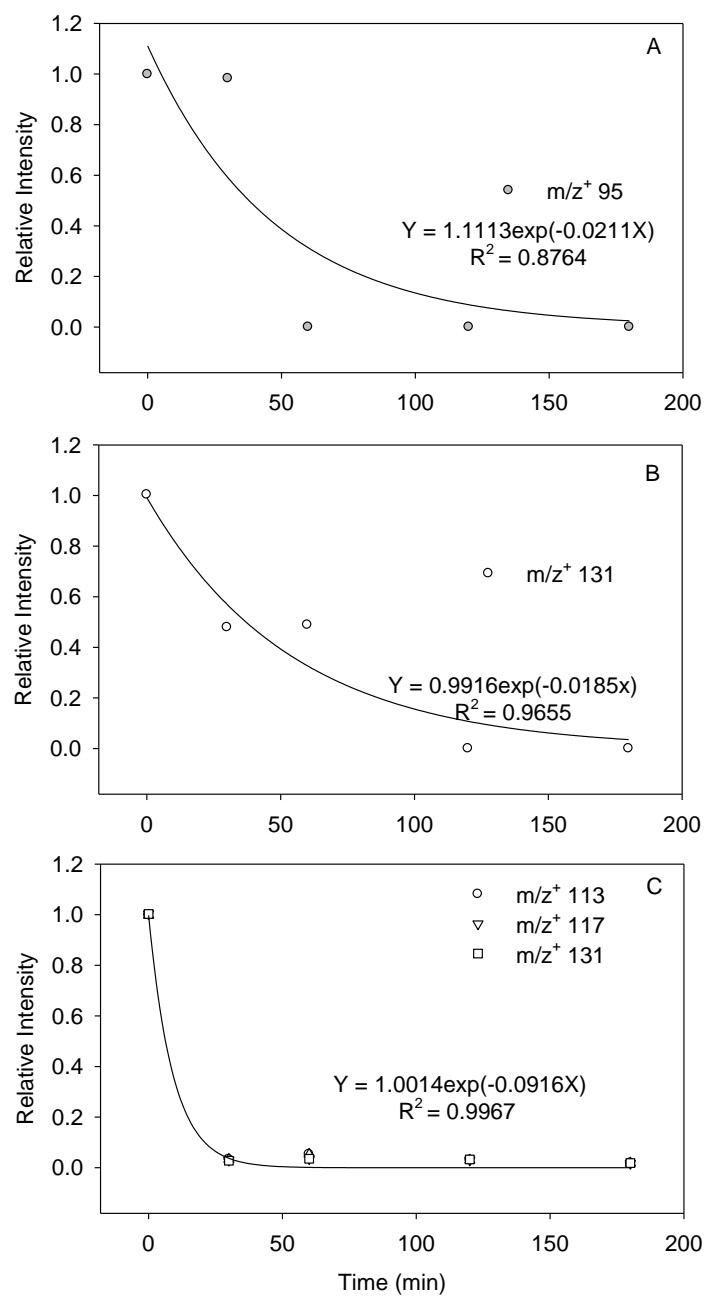

**Figure 3: The decay of glyoxal in SA aerosols in the humid (A) and in the dry chamber (B), and in AS aerosols in the dry chamber (C)**



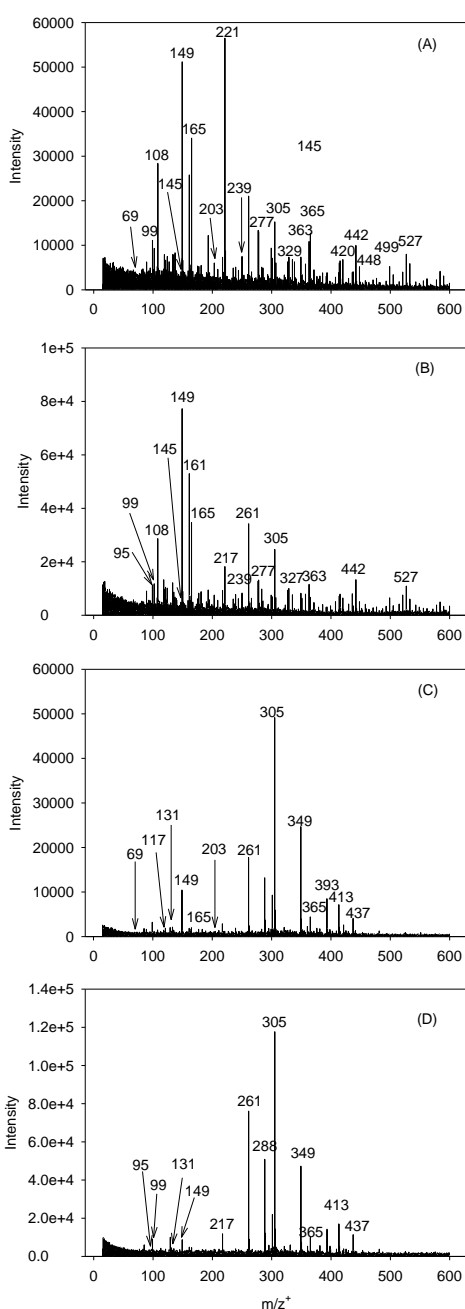

**Figure 4: Positive-mode mass spectra for 3 hour-irradiated AS aerosols (A) and SA aerosols (B) in the humid chamber, and AS aerosols (C) and SA aerosols (D) in the dry chamber**





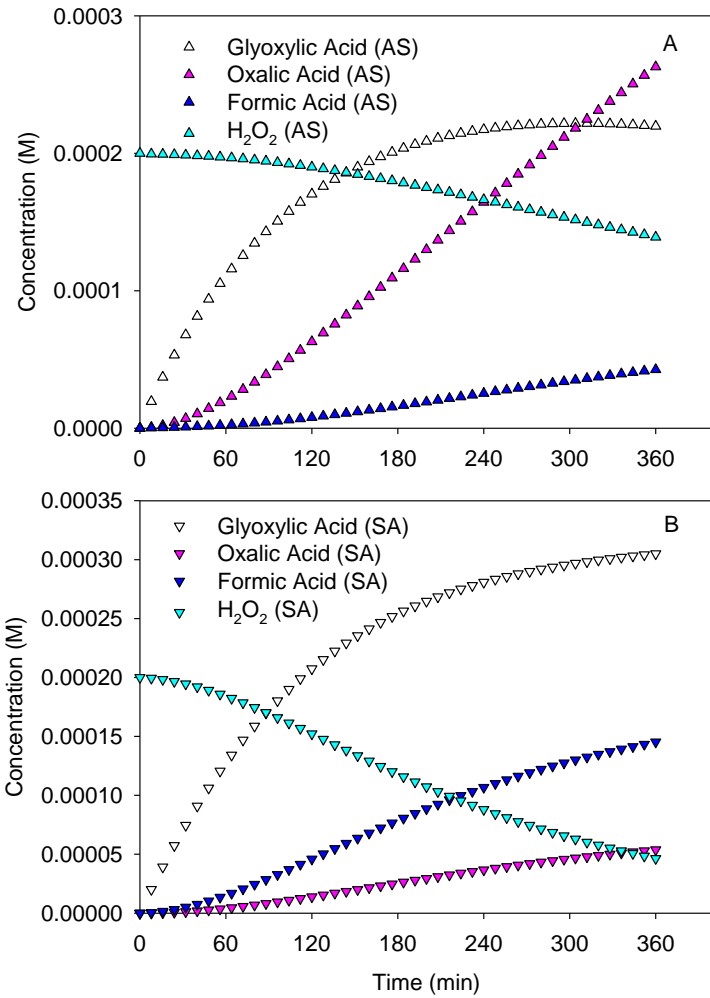

**Figure 5: Simulated concentrations in ALW of AS aerosols (A) and SA aerosols (B) during 3 hour irradiation in the humid chamber**





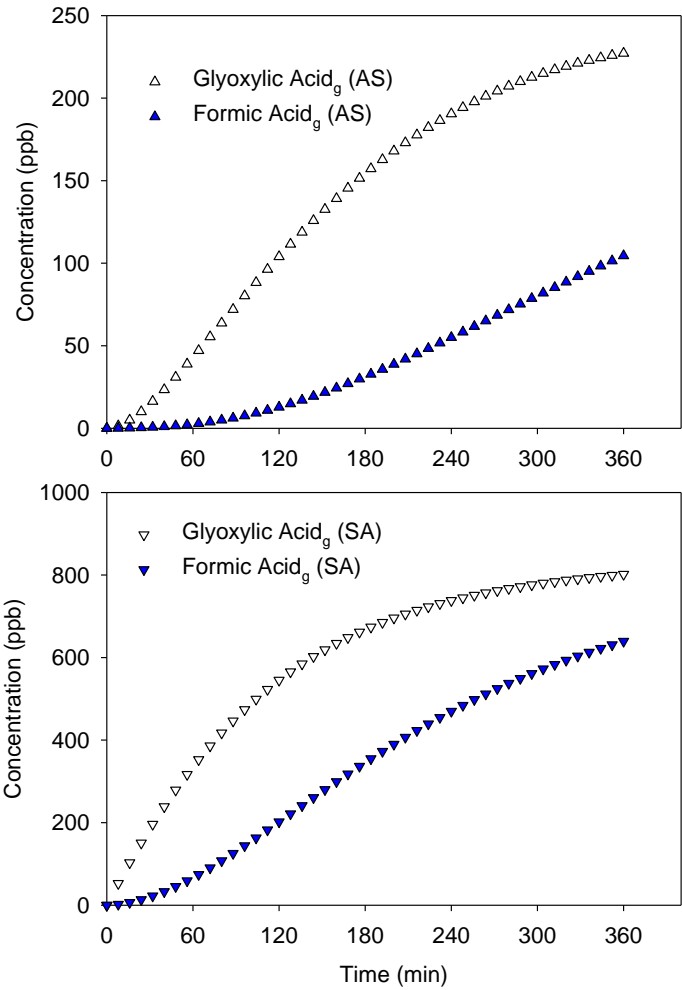

**Figure 6:  Gas-phase simulations of glyoxylic acid and formic acid evaporated from AS aerosols (A) and SA aerosols (B) during 3 hour irradiation in the humid chamber**