# Peer review of "Photochemical Organonitrate Formation in Wet Aerosols"

_Atmospheric Chemistry and Physics, 2016_

## Referee Comment (RC1) · Anonymous Referee #2 · 30 May 2016

**General Comments**

This paper presents evidence of formation of organonitrates, organosulfates, and mixed nitrate-sulfate organic compounds from glyoxal. Their formation is attributed to aqueous phase processing of glyoxal and its hydrated forms. While photochemistry of gas-phase species enhanced the formation of the organitrogen and organosulfar compounds, the authors present reasonable evidence that this was due to enhanced formation of HNO3, which partitioned into aerosol and enhanced aqueous processing. Also, the products were formed during photochemical experiments were also formed without UV irradiation. An existing model of aqueous-aerosol glyoxal chemistry was modified to include some new reactions and partitioning of glyoxal. The formation of these interesting organitrogen and organosulfur compounds seems clear, supported by identification with mass spectrometry and a fairly straightforward experimental design.

[Figure]

The data to support conclusions regarding kinetics of these reactions and subsequent modeling is limited, and it is not clear if any strong conclusions can be made by comparison with a kinetics model. The main result of this work is the identification of the products, with the potential of their formation in atmospheric aerosol via the aqueous chemistry presented here. No attempt was made to track the total amount of oxidized forms of nitrogen (e.g. NOx, organonitrates, HNO3), and this should be done during revision. For example, can the observed changes in gas-phase NOx levels be reasonably attributed to known sinks? The lack of detection of glyoxal or its hydrated forms in humidified ammonium sulfate aerosol, even at the beginning of the experiment, is somewhat puzzling and must be explained further. This work is significant in the identification of formation of organitrogen and organosulfur compounds from glyoxal chemsitry. Therefore I recommend this work for publication, pending revisions. Fundamental points still need to be addressed, and a number of clarifications are required prior to publication, as detailed below.

Specific Comments

(2, 39) It should be noted that the low volatility of glyoxal results largely from the high level of hydration that occurs upon dissolution in water.

(4, 6) It is has been shown that drying can induce chemistry in aqueous aerosols.(1-3) The aerosol in this study contained glyoxal prior to drying and addition to the chamber. Were there any indications that chemistry occurred during that drying process?

(4, 15) The humidifying process should be described in further detail. It is currently described as the chamber being filled with clean dry air and then humidified. It doesn't seem feasible that 90% RH can be reached with the chamber initially full of dry air.

(4, 12) Please elaborate on the relevance of your gas and particle phase concentrations to the atmosphere. Although the goal of this study is largely to show the potential source of these compounds and the link to aqueous processing, the relation to the atmospheric conditions should be addressed further.

(4, 23) The use of E-AIM will also provide, as you note, the pH of the aerosol, yet pH is not reported here. pH will affect particle equilibria, partitioning, and may change the resulting chemistry. It is certainly an important environmental variable that should be reported for all experiments in Table S1. A general comment on pH and potential effects should be included in your updated discussion, particularly since acidity was a major aspect of your experiments (sulfuric acid seed vs. ammonium sulfate seed).

(5, 13-19) This section has the heading "... and 226", but no mention is made of m/z 226.

(6, 12-15) The authors observed that experiments that are similar, except for the presence of glyoxal (#2 and #7), had very different NOx chemistry, but do not explain this. The sinks and consequences of gas-phase NOx should be more clearly discussed, particularly in light of your observations. For example, if NOx is converted to HNO3 and partitions to aerosol, pH could be significantly altered.

(7, 9-11) Were NO2+ to be formed in any significant amount, would this now be a potentially important reactive species (nucleophile) in your aerosol? Are there any indications that this is the case?

(8, 15) The authors state that no glyoxal peak was observed in mass spectra for the humid chamber AS aerosols, yet you do observe organonitrate products (Fig. 1). Does your model suggest complete and rapid conversion of glyoxal to products? Given the importance of ALW for partitioning of glyoxal, it is puzzling that AS aerosol under humid conditions does not contain glyoxal. This important point was dismissed by the authors.

(9, 32) The aerosols evaporate to maintain equilibrium at the RH conditions of the chamber, not due to surface area considerations. What comment about surface area was intended?

Technical Comments

(Page 1, Line 13) change to read "or sulfuric acid particles"

(Page 1, Line 33) This sentence is awkward, but it makes an important point that SOA(aq) is likely to improve model predictions. Please make this sentence clearer, perhaps split into two.

(2, 12) Add references 4 and 5.

(2,18) Add reference 6.

(2, 20) change to read "..compounds, OH radicals, and water.."

(3, 6) The importance of more realistic aerosol composition should be noted here. Ambient aerosol will have a wide range of organic compounds in addition to those derived from glyoxal.(Refs 7,8) Recent work suggests that compounds like glyoxal will from condensation products (acetals, etc.) with these other aerosol constituents.(Ref 9) This could affect the chemistry studied in this work, by reducing the amount of glyoxal available for reaction and potentially changing the product distribution. The authors should address the effect of actual ambient aerosol composition.

(3, 23) remove the first word : "the"

(3, 35) change to read ".., liquid water, and . . ."

(5, 30-31) change to read ". . .08C11) and not likely nitric acid adducts.."

(5, 37) change to read "..Cole, 2000), and MIDAS does not propose.."

(6, 12-13) change to read "Experiment #2) also shows significant.."

(6, 16) This is an interesting style of using an introductory question. It would be better to use a direct statement rather than giving the reader some suspense. Ambiguity impedes clarity. Please rephrase as a direct statement, such as "Aqueous phase chemistry and photochemistry may lead to volatile products that contribute to gas phase peroxy radicals"

(6, 37) and (7, 1) and throughout the manuscript, change to read ". . .after 3 hours of

irradiation. . .”

(7, 3-5) While heterogeneous reactions are a possible source, do the authors consider OH + NO2 a source of HNO3? Is this included in the model?

(7, 12) change to read “Figure 2 suggests. . .”

(8, 3-4) change to read “During irradiation, oxalic acid was formed in the humid chamber, shown by UPLC-Q-TOF-MS detection of m/z- 89. . .”

(9,28) through (10, 29) The time-resolved data should be addressed within the context of other studies. Particularly for the reduced nitrogen species (imines, imidizoles, etc.) Studies have looked at this reaction under a wide range of conditions, which should allow comparison.(Refs 10-12)

(10, 18) change to read “. . . because aqueous phase reactions of glyoxal with ammonium form imines...”

(10, 22-23) change to read “. . .form oligomers and imines. In SA aerosols the formation. . .”

(11, 8) change to read “. . . during the daytime. Notably, nitrate concentrations. . .” Figures

Scheme S1. This should be placed into the main manuscript. You discuss extensively the formation of these organonitrates, so this should not be supplemental.

Figure 1. This figure should be a 4 panel grid, with the spectra for humid conditions on the top row, and dry conditions on the bottom row, with AS results on the left and SA results on the right. It is difficult to compare in a single column. Each figure (a-d) should have a label denoting the aerosol type and the humidity level.

Figure 3. The legends must be moved to the top right corner to avoid confusion between the data and the legend. R-squared should be reported to at most 3 decimal places (0.001). The linear fits do not need to fully displayed, only the time constants.

The linear fit intercepts should all be 1.0. Instead of presenting the equations, you should label the plots with the effective lifetime or the half-life of the glyoxal.

Figure 4. The same 4-panel grid format as suggested for Figure 1 should be used.

Bibliography

1. Nguyen, T. B. et al. Formation of nitrogen- and sulfur-containing light-absorbing compounds accelerated by evaporation of water from secondary organic aerosols. J. Geophys. Res. 117, 1–14 (2012).

2. Laskin, A., Laskin, J. & Nizkorodov, S. a. Chemistry of Atmospheric Brown Carbon. Chem. Rev. 115, (2015).

3. De Haan, D. O. et al. Secondary organic aerosol formation by self-reactions of methylglyoxal and glyoxal in evaporating droplets. Environ. Sci. Technol. 43, 8184–90 (2009).

4. Liggio, J. Reactive uptake of glyoxal by particulate matter. J. Geophys. Res. 110, 1–13 (2005). 5. Corrigan, A. L., Hanley, S. W. & De Haan, D. O. Uptake of Glyoxal by Organic and Inorganic Aerosol. Environ. Sci. Technol. 42, 4428–4433 (2008).

6. Drozd, G., Woo, J., Häkkinen, S. A. K., Nenes, A. & McNeill, V. F. Inorganic salts interact with oxalic acid in submicron particles to form material with low hygroscopicity and volatility. Atmos. Chem. Phys. 14, 5205–5215 (2014).

7. Goldstein, A. H. & Galbally, I. E. Known and Unexplored Organic Constituents in the Earth's Atmosphere. Environ. Sci. Technol. 41, 1514–1521 (2007).

8. Hallquist, M. et al. The formation, properties and impact of secondary organic aerosol: current and emerging issues. Atmos. Chem. Phys. 9, 5155–5236 (2009).

9. Drozd, G. T. & McNeill, V. F. Organic matrix effects on the formation of light-absorbing compounds from $\alpha$-dicarbonyls in aqueous salt solution. Environ. Sci. Process. Impacts 16, 741–747 (2014).
10. Yu, G. et al. Glyoxal in aqueous ammonium sulfate solutions: products, kinetics and hydration effects. Environ. Sci. Technol. 45, 6336–42 (2011).

11. Sareen, N., Schwier, A. N., Shapiro, E. L., Mitroo, D. & McNeill, V. F. Secondary organic material formed by methylglyoxal in aqueous aerosol mimics. Atmos. Chem. Phys. 10, 997–1016 (2010).

12. Nozière, B., Dziedzic, P. & Córdova, A. Products and kinetics of the liquid-phase reaction of glyoxal catalyzed by ammonium ions (NH4(+)). J. Phys. Chem. A 113, 231–7 (2009).

---

## Referee Comment (RC2) · Anonymous Referee #1 · 11 Jun 2016

In this paper the authors describe results of environmental chamber experiments in which they investigate the formation of organic nitrates in aqueous aerosol from reaction of glyoxal with OH radicals under high NOx conditions. Experiments were also conducted in the dark with O3 and probably NO3 radicals and in the absence of oxidants. Aerosol products were collected on filters and analyzed offline by electrospray mass spectrometry to obtain elemental formulas that were used to assign compound identity.

The experiments are well done and the discussion of results is very thorough and reasonable, although I have concerns about proposed product assignments and reaction mechanisms, as noted below. The results have the potential to be important, since organic nitrates are an important class of compounds. Determining the extent to which those present in the atmosphere are formed through aqueous chemistry matters because whereas gas phase formation from RO2 + NO reactions sequester NOx and radicals, aqueous phase formation from aldehydes/alcohols and nitric acid does not. I think the paper may be suitable for publication in ACP but have some significant comments that should first be addressed.

Specific Comments

1. Page 4, lines 28-29: It is well known that sonication in water can form hydrogen peroxide and possibly other oxidants. What tests were conducted to ensure that this did not affect the composition of the samples?

2. Page 5, line 7: 150 ppm uncertainty in mass assignments seems large. I am used to values less than 5 ppm for accurate assignments of elemental formulas. Why is this, and how does this affect the assignment of elemental formulas? For example, in Tables 1-3 simple molecules like sulfuric acid and its dimer can be identified at the sub-5 ppm level, but the proposed organic compounds cannot. This seems problematic. This issue should be discussed, and the authors should show what other possible products can be assigned with similar or better uncertainties. As it is, I do not have much confidence in these assignments, including the organic nitrates that are the focus of the paper.

3. Page 5, lines 32-33: It is difficult to believe that an alcohol would be converted to an organic nitrate in aqueous solution by a reversible reaction (the single arrow shown should be a double arrow for equilibrium). I would expect the water to shift the product distribution fully towards the alcohol. If such chemistry can occur it should be known for polyols or at least simple secondary alcohols, so literature should be cited and discussed to support these speculations. Although the mechanism provides an explanation for these particular products, it seems that one would expect many other organic nitrates from this mechanism as well. Why only these two out of many possibilities? This needs more discussion.

4. Page 7, line 33: Should "hydroxide" be "nitrate"? Where would one find ammonium

hydroxide in aerosols?

5. Page 10, line 14: What is meant by "deprotonated acids"? Do you mean carboxylate ions? Ions do not evaporate from solutions.

6. Page 11, lines 5-11: Should be more specific in this paragraph. Do you mean total nitrates or inorganic nitrates or organic nitrates? Organic nitrates are not very soluble in water unless they are low molecular weight and multifunctional.

7. It is well established that electrospray ionization is highly sensitive to compound structure and the sample matrix. How is it known that the signals assigned to organic nitrates are not just trace components that are not a significant portion of the aerosol mass? Some discussion of quantitation is warranted.

8. Since the major conclusion of this paper appears to be that organic nitrates can be formed by mixing glyoxal and nitric acid in water in the absence of oxidants, the authors should better explain the point of the environmental chamber experiments. Showing that nitric acid can be formed under these photooxidation conditions seems unimportant, since it is readily formed in the atmosphere. How do the chamber results contribute to the conclusions?

Technical Comments

1. Page 1, line 27: "aerosols" should probably be "products"

2. Page 2, line 36: "ethene" should be "acetylene"

3. Page 3, line 8: "chromatography" should be "chromatograph"

4. Page 5, line 26: "nitrate" should be "nitric"

5. Page 9, line 20: Should add "area" after "surface"

6. Page 9, line 38: "catalyzation" should be "catalysis"

---

## Author Comment (AC1) · 10 Aug 2016

We thank the reviewers for careful reading and helpful comments that improve the quality of the manuscript.    Reviewer comments have been copied followed by our responses in bold.

**R1 (Anonymous Referee #2)**

General Comments
This paper presents evidence of formation of organonitrates, organosulfates, and mixed nitrate-sulfate organic compounds from glyoxal. Their formation is attributed to aqueous phase processing of glyoxal and its hydrated forms. While photochemistry of gas-phase species enhanced the formation of the organitrogen and organosulfar compounds, the authors present reasonable evidence that this was due to enhanced formation of HNO3, which partitioned into aerosol and enhanced aqueous processing.
Also, the products were formed during photochemical experiments were also formed without UV irradiation. An existing model of aqueous-aerosol glyoxal chemistry was modified to include some new reactions and partitioning of glyoxal. The formation of these interesting organitrogen and organosulfur compounds seems clear, supported by identification with mass spectrometry and a fairly straightforward experimental design.
The data to support conclusions regarding kinetics of these reactions and subsequent modeling is limited, and it is not clear if any strong conclusions can be made by comparison with a kinetics model. The main result of this work is the identification of the products, with the potential of their formation in atmospheric aerosol via the aqueous chemistry presented here.

R1C1) No attempt was made to track the total amount of oxidized forms of nitrogen (e.g. NOx, organonitrates, HNO3), and this should be done during revision. For example, can the observed changes in gas-phase NOx levels be reasonably attributed to known sinks?

**Response) We did not attempt to measure $HNO_3$ uptake quantitatively. A FACSIMILE model predicts ~500 $\mu$M of $HNO_3$ uptake in the aqueous phase after 3 hour irradiation when initial conditions are 10 ppb of NO, 0 ppb of $NO_2$, 10 ppb of $O_3$ and 500 ppb of an organic compound in the gas phase (glyoxylic acid). This concentration of $HNO_3$ is comparable to that of inorganic constituents in wet aerosols (200 $\mu$M of ammonium sulfate/sulfuric acid). We added this point at the end of section 3.3.**

**"We also estimate the concentration of $HNO_3$ taken up into wet aerosols by including chemistry of $NO_x$, $HO_x$, peroxy radical, $HNO_3$ partitioning into a FACSIMILE model. ~ 500 $\mu$M of $HNO_3$ uptake in the aqueous phase is predicted after 3 hours of irradiation when initial conditions are 10 ppb of NO, 0 ppb of $NO_2$, 10 ppb of $O_3$ and 500 ppb of an organic compound in the gas phase (glyoxylic acid). This concentration of $HNO_3$ is sufficient to form organonitrates with glyoxal and comparable to that of other inorganic constituents in wet aerosols (200 $\mu$M of ammonium sulfate/sulfuric acid)."**

R1C2) The lack of detection of glyoxal or its hydrated forms in humidified ammonium sulfate aerosol, even at the beginning of the experiment, is somewhat puzzling and must be explained further.

**Response) Thank you for pointing this out. We reinvestigated and found that $m/z^+$ 131 ([glyoxal + $2H_2O$ + Na]$^+$) was actually detected in 15 and 30 minute samples. But $m/z^+$ 131 that we detected in 0 minute sample was an imine (C4H7N2O3), which appeared to overlap the glyoxal peak. So, a decay plot cannot be constructed.**

**We changed the sentence in the paragraph (line 15, page 8) as follows:**
**"(glyoxal was also detected for AS aerosols in the humid chamber; however, since it was only detected in 15 and 30 minute samples, no decay plot was constructed)"**

**We also added the following in the text (line 12, page 8):**

**"131 (= [M + H$_2$O + MeOH + Na]$^+$)"**

This work is significant in the identification of formation of organitrogen and organosulfur compounds from glyoxal chemsitry. Therefore I recommend this work for publication, pending revisions. Fundamental points still need to be addressed, and a number of clarifications are required prior to publication, as detailed below.

Specific Comments
R1C3) (2, 39) It should be noted that the low volatility of glyoxal results largely from the high level of hydration that occurs upon dissolution in water.

**Response) Clearly glyoxal undergoes hydration, but immediately hydrated glyoxal forms oligomers via acid catalysis and organic-inorganic complexes in the presence of inorganic constituents. These are likely to be SOA from glyoxal. And these points were already addressed in the previous paragraph (line 10-18, page 2).**

**We have modified that sentence to clarify the reviewer's point:**
**"Water soluble organic compounds like glyoxal and methylglyoxal hydrate and form oligomers through hemiacetal formation and aldol condensation, especially in evaporating droplets."**

R1C4) (4, 6) It is has been shown that drying can induce chemistry in aqueous aerosols. (1-3) The aerosol in this study contained glyoxal prior to drying and addition to the chamber. Were there any indications that chemistry occurred during that drying process?

**Response) Yes. In Fig. S6, solutions and aerosols in the chamber at 0 minute were directly compared. Imines and acid-catalyzed oligomers were found in the aerosols. This is discussed in the text (line 13, page 10).**

R1C5) (4, 15) The humidifying process should be described in further detail. It is currently described as the chamber being filled with clean dry air and then humidified. It doesn't seem feasible that 90% RH can be reached with the chamber initially full of dry air.

**Response) This humidifier was developed by modifying an existing commercial humidifier. Water spray and evaporation pan were specially designed to generate water steam rapidly. While adding water vapors into the smog chamber, we monitored SMPS to ensure that no water droplet was introduced. This device will be requested for patent in the future, so we cannot discuss more in details.**

**We added:**
**"This humidifier was developed by modifying an existing commercial humidifier. Water spray and evaporation pan were specially designed to generate water steam rapidly. While adding water vapors into the chamber, we monitored SMPS to ensure no water droplet was introduced."**

R1C6) (4, 12) Please elaborate on the relevance of your gas and particle phase concentrations to the atmosphere. Although the goal of this study is largely to show the potential source of these compounds and the link to aqueous processing, the relation to the atmospheric conditions should be addressed further.

**Response) We added the following:**
**"Concentrations of NO$_x$, O$_3$, and particle mass in smog chamber can be related to a moderate haze condition in urban areas, particularly observed in Seoul or Beijing."**

R1C7) (4, 23) The use of E-AIM will also provide, as you note, the pH of the aerosol, yet pH is not reported here. pH will affect particle equilibria, partitioning, and may change the resulting chemistry. It is certainly an important environmental variable that should be reported for all experiments in Table S1. A general comment on pH and potential effects should be included in your updated discussion, particularly since acidity was a major aspect of your experiments (sulfuric acid seed vs. ammonium sulfate seed).

**Response) We have added pH values in Table S1. It appears acidity affects oligomerization in SA aerosols, and this is discussed in line 22-29 on page 10.**

R1C8) (5, 13-19) This section has the heading ": : : and 226", but no mention is made of m/z 226.

**Response) We change the section title to:**
**"MS/MS analysis for m/z⁻ 147 and standard MS analysis for m/z⁻ 147 and 226"**

R1C9) (6, 12-15) The authors observed that experiments that are similar, except for the presence of glyoxal (#2 and #7), had very different NOx chemistry, but do not explain this. The sinks and consequences of gas-phase NOx should be more clearly discussed, particularly in light of your observations. For example, if NOx is converted to HNO3 and partitions to aerosol, pH could be significantly altered.

**Response) Please see our response for R1C1 regarding sinks of NOx to HNO3. NO2 is effectively formed by peroxy radical-NO reactions (#2), but is only slightly increased due to the lack of peroxy radicals when there is no glyoxal in the beginning (#7). This point is already discussed in line 10 on page 6. There was no evidence of enhanced oligomer formation by HNO3 uptake while more oligomers were observed in SA aerosols. We modified the text (line 3, page 10):**

**"However, acidity effects on oligomer formation requires further study because sulfuric acid in SA aerosols appears to enhance oligomerization while photochemically formed nitric acid does not."**

R1C10) (7, 9-11) Were NO2+ to be formed in any significant amount, would this now be a potentially important reactive species (nucleophile) in your aerosol? Are there any indications that this is the case?

**Response) We did not attempt to measure $NO_2^+$, so heterogeneous reactions of $NO_2^+$ is beyond the scope. In this paper, we focus on nitrates and organonitrates. We just mention this since no $HNO_3$ was observed in SA aerosols.**

**We have now added:**
**"If $NO_2^+$ were formed in a significant amount, it could be an important reactive species. However, measurement of $NO_2^+$ and investigation of its potential role is beyond the scope of this study."**

R1C11) (8, 15) The authors state that no glyoxal peak was observed in mass spectra for the humid chamber AS aerosols, yet you do observe organonitrate products (Fig. 1). Does your model suggest complete and rapid conversion of glyoxal to products? Given the importance of ALW for partitioning of glyoxal, it is puzzling that AS aerosol under humid conditions does not contain glyoxal. This important point was dismissed by the authors.

**Response) See our response for R1C2.**

R1C12) (9, 32) The aerosols evaporate to maintain equilibrium at the RH conditions of the chamber, not due to surface area considerations. What comment about surface area was intended?

**Response) This has been changed in the text to:**
**"When the solutions are atomized and introduced into the smog chamber, water evaporates to equilibrate to the chamber RH, and concentration of solutes increase."**

Technical Comments
R1C13) (Page 1, Line 13) change to read "or sulfuric acid particles"

**Response) It was already written as "or sulfuric acid particles."**

(Page 1, Line 33) This sentence is awkward, but it makes an important point that SOA(aq) is likely to improve model predictions. Please make this sentence clearer, perhaps split into two.

**Response) Now it reads:**

**"Including SOAaq is likely to improve model predictions, which currently underestimate the ambient concentration and oxidation state of organic aerosols. Water soluble organic compounds with a small carbon number (C2-C3) are not considered precursors to SOA formation through gas-phase chemistry and vapor pressure driven partitioning (Pankow, 1994) because of their high vapor pressure. However, they are potential SOAaq precursors."**

R1C14) (2, 12) Add references 4 and 5.

**Response) We add suggested references.**

R1C15) (2,18) Add reference 6.

**Response) We add suggested references.**

R1C16) (2, 20) change to read "..compounds, OH radicals, and water.."

**Response) Now it reads as the reviewer suggests.**

R1C17) (3, 6) The importance of more realistic aerosol composition should be noted here. Ambient aerosol will have a wide range of organic compounds in addition to those derived from glyoxal.(Refs 7,8) Recent work suggests that compounds like glyoxal will from condensation products (acetals, etc.) with these other aerosol constituents.(Ref 9)
This could affect the chemistry studied in this work, by reducing the amount of glyoxal available for reaction and potentially changing the product distribution. The authors should address the effect of actual ambient aerosol composition.

**Response) Glyoxal is a surrogate of water soluble organic compounds in ambient wet aerosols. Radical and non-radical reactions of water soluble organic compounds in wet aerosols are expected to be seen in our glyoxal reactions. Glyoxal undergoes self-oligomerization through hemiacetal formation and aldol condensation leading to light absorbing products (Shapiro et al., 2009). Ammoniums, sulfates and nitrates are main inorganic constituents (Zhang et al., 2007), and glyoxal reacts with them. OH reactions of glyoxal produce dicarboxylic acids (e.g., oxalic acid), which are also expected to be the products of water soluble organic compounds. These non-radical reactions compete with OH reactions. In our smog chamber experiments, non-radical reactions are more dominant than OH reactions in the condensed phase and we expect this is true for water soluble organic compounds in ambient wet aerosols.**

**We added the following in the sentence:**
**"glyoxal is used, as a surrogate of water soluble organic compounds in ambient wet aerosols, to explore non-radical and radical reactions in the condensed phase leading to SOA."**

R1C18) (3, 23) remove the first word : "the"

**Response) Now "the" is removed.**

R1C19) (3, 35) change to read ".., liquid water, and : : :"

**Response) We do not understand this request. Perhaps the page or line number the reviewer is referring to is incorrectly written??**

R1C20) (5, 30-31) change to read ": : :08C11) and not likely nitric acid adducts.."

**Response) Now "and" is inserted**

R1C21) (5, 37) change to read "..Cole, 2000), and MIDAS does not propose.."

**Response) Now "while" is changed to "and."**

R1C22) (6, 12-13) change to read "Experiment #2) also shows significant.."

**Response) Now "the" is removed, as suggested.**

R1C23) (6, 16) This is an interesting style of using an introductory question. It would be better to use a direct statement rather than giving the reader some suspense. Ambiguity impedes clarity. Please rephrase as a direct statement, such as "Aqueous phase chemistry and photochemistry may lead to volatile products that contribute to gas phase peroxy radicals"

**Response) Now it reads as the review suggests:**
**"Photochemistry on wet aerosols may lead to volatile organic products that contribute to gas-phase peroxy radicals."**

R1C24) (6, 37) and (7, 1) and throughout the manuscript, change to read ": : :after 3 hours of irradiation: : :"

**Response) Now we changed according to the reviewer's suggestion.**

R1C25) (7, 3-5) While heterogeneous reactions are a possible source, do the authors consider OH + NO2 a source of HNO3? Is this included in the model?

**Response) We include this as an aqueous phase reaction. NO$_2$ concentration in the aqueous phase is determined by the Henry's law constant. The contribution of this reaction to HNO$_3$ is, however, small.**

R1C26) (7, 12) change to read "Figure 2 suggests: : :"

**Response) We removed "also" to read as suggested.**

R1C27) (8, 3-4) change to read "During irradiation, oxalic acid was formed in the humid chamber,

shown by UPLC-Q-TOF-MS detection of m/z- 89: : :"

**Response) We changed according to the reviewer's suggestion.**

R1C28) (9,28) through (10, 29) The time-resolved data should be addressed within the context of other studies. Particularly for the reduced nitrogen species (imines, imidizoles, etc.) Studies have looked at this reaction under a wide range of conditions, which should allow comparison.(Refs 10-12)

**Response) We already discussed previous studies of nitrogen-containing organics (line 13-15, page 2). So we have added the suggested references (10-12) there.**

R1C29) (10, 18) change to read ": : : because aqueous phase reactions of glyoxal with ammonium form imines..."

**Response) Now we remove "in the" to read as suggested.**

R1C30) (10, 22-23) change to read ": : :form oligomers and imines. In SA aerosols the formation: : :"

**Response) We split the sentence into two as the reviewer suggests.**

R1C31) (11, 8) change to read ": : : during the daytime. Notably, nitrate concentrations: : :" Figures

**Response) Now we removed "And," as suggested**

R1C32) Scheme S1. This should be placed into the main manuscript. You discuss extensively the formation of these organonitrates, so this should not be supplemental.

**Response) We agree. Now Scheme S1 is Scheme 4. The previous Scheme 4 is now Scheme 5.**

R1C33) Figure 1. This figure should be a 4 panel grid, with the spectra for humid conditions on the top row, and dry conditions on the bottom row, with AS results on the left and SA results on the right. It is difficult to compare in a single column. Each figure (a-d) should have a label denoting the aerosol type and the humidity level.

**Response) Figure 1 is now a 4 panel grid as the reviewer suggests.**

R1C34) Figure 3. The legends must be moved to the top right corner to avoid confusion between the data and the legend. R-squared should be reported to at most 3 decimal places (0.001). The linear fits do not need to fully displayed, only the time constants. The linear fit intercepts should all be 1.0. Instead of presenting the equations, you should label the plots with the effective lifetime or the half-life of the glyoxal.

**Response) Legends are now placed on the top right corner. Instead of the equations, the rate constants, k (min-1), are used. R-squared has now 3 decimal places.   Y-intercept value (at t = 0) for A is now 1.**

R1C35) Figure 4. The same 4-panel grid format as suggested for Figure 1 should be used.

**Response) Figure 4 is now a 4 panel grid as the reviewer suggests.**

---

## Author Comment (AC2) · 10 Aug 2016

We thank the reviewers for careful reading and helpful comments that improve the quality of the manuscript.   Reviewer comments have been copied followed by our responses in bold.

**R2 (Anonymous Referee #1)**

In this paper the authors describe results of environmental chamber experiments in which they investigate the formation of organic nitrates in aqueous aerosol from reaction of glyoxal with OH radicals under high NOx conditions. Experiments were also conducted in the dark with O3 and probably NO3 radicals and in the absence of oxidants. Aerosol products were collected on filters and analyzed offline by electrospray mass spectrometry to obtain elemental formulas that were used to assign compound identity.
The experiments are well done and the discussion of results is very thorough and reasonable, although I have concerns about proposed product assignments and reaction mechanisms, as noted below. The results have the potential to be important, since organic nitrates are an important class of compounds. Determining the extent to which those present in the atmosphere are formed through aqueous chemistry matters because whereas gas phase formation from RO2 + NO reactions sequester NOx and radicals, aqueous phase formation from aldehydes/alcohols and nitric acid does not. I think the paper may be suitable for publication in ACP but have some significant comments that should first be addressed.

Specific Comments
R2C1) Page 4, lines 28-29: It is well known that sonication in water can form hydrogen peroxide and possibly other oxidants. What tests were conducted to ensure that this did not affect the composition of the samples?

**Response) In our previous reaction vessel experiments, we used excess hydrogen peroxide as an OH radical source. Addition of hydrogen peroxide to standards verified that hydrogen peroxide does not react with glyoxal or oxalic acid. It does oxidize glyoxylic acid (Tan et al., 2009). However, in this smog chamber experiments, glyoxylic acid evaporates and becomes a major source of peroxy radicals during the irradiation (line 30, page 6). Therefore, a small amount of hydrogen peroxide formed during sonication is very unlikely to affect the sample composition. We add the following:**

**"Note that any possible hydrogen peroxide formed during sonication is not likely to affect aqueous-phase photooxidation. According to our previous reaction vessel experiments (Tan et al., 2009), hydrogen peroxide does not react with glyoxal or oxalic acid. It only oxidizes glyoxylic acid. But in this smog chamber experiments, glyoxylic acid evaporates."**

R2C2) Page 5, line 7: 150 ppm uncertainty in mass assignments seems large. I am used to values less than 5 ppm for accurate assignments of elemental formulas. Why is this, and how does this affect the assignment of elemental formulas? For example, in Tables 1-3 simple molecules like sulfuric acid and its dimer can be identified at the sub- 5 ppm level, but the proposed organic compounds cannot. This seems problematic. This issue should be discussed, and the authors should show what other possible products can be assigned with similar or better uncertainties. As it is, I do not have much confidence in these assignments, including the organic nitrates that are the focus of the paper.

**Response) Unlike FTICR-MS, whose uncertainty is below 1 ppm, TOF-MS tends to have uncertainties. Actually this is not surprising because it is well known that uncertainties of TOF could be high (Smith et al., 2013). Note that throughout this work, if MIDAS provided more than one possible chemical formula, we selected the one with the lowest ppm difference. Regarding organonitrates, we also conducted an MS/MS analysis for m/z $^-$ 147, which clearly indicated organonitrates by showing a nitrate fragment (m/z $^-$ 62) (Fig. S2) and an ESI-MS analysis for a standard mixture solution of glyoxal and nitric acid, confirming m/z $^-$ 147 and 226 were organonitrates (Fig. S3). Structures and mechanisms are proposed based on chemical**

**formula provided by MIDAS with the lowest ppm difference.**

R2C3) Page 5, lines 32-33: It is difficult to believe that an alcohol would be converted to an organic nitrate in aqueous solution by a reversible reaction (the single arrow shown should be a double arrow for equilibrium). I would expect the water to shift the product distribution fully towards the alcohol. If such chemistry can occur it should be known for polyols or at least simple secondary alcohols, so literature should be cited and discussed to support these speculations. Although the mechanism provides an explanation for these particular products, it seems that one would expect many other organic nitrates from this mechanism as well. Why only these two out of many possibilities? This needs more discussion.

**Response) We provided references for the formation of organonitrates (nitrate esters). As we all know, alcohols and sulfuric acid form organosulfates (sulfate esters). Esterification for organonitrate and organosulfate formation seems to be more favored than hydrolysis in the condensed phase (aerosol phase). Even in solutions we observed organosulfate formation (m/z-217), as shown in Table 3. Yes, organonitrate formation suggested in Scheme S1 is a proposed one. There could be other ways of forming organonitrates; however, the chemical formula provided is the one with the lowest ppm difference suggested by MIDAS.**

**We have provided references, and we now state:**
**"The proposed formation and molecular structures are illustrated in Scheme S1. Other organonitrates may form through this mechanism as well. However, the chemical formula provided herein has the smallest error compared to the measured mass."**

**We replaced a single arrow by a double arrow.**

R2C4) Page 7, line 33: Should "hydroxide" be "nitrate"? Where would one find ammonium hydroxide in aerosols?

**Response) Ammonium hydroxide was used in our previous experiments (Ortiz-Montalvo et al., 2014; Tan et al., 2009). To clarify, we have modified this sentence:**
**"While sulfuric acid and ammonium hydroxide addition do not interfere the real-time formation of oxalic acid in dilute (cloud-relevant) photooxidation experiments…"**

R2C5) Page 10, line 14: What is meant by "deprotonated acids"? Do you mean carboxylate ions? Ions do not evaporate from solutions.

**Response) Page 10 should be page 9. We changed "deprotonated acids" to "undissociated acids."**

R2C6) Page 11, lines 5-11: Should be more specific in this paragraph. Do you mean total nitrates or inorganic nitrates or organic nitrates? Organic nitrates are not very soluble in water unless they are low molecular weight and multifunctional.

**Response) We mean nitrates. They are major constituents, and form organonitrates in aerosols. They facilitate water uptakes. Alkylnitrates (= organic nitrates or organonitrates) formed in the gas phase were already discussed in line 32-38, page 10. And we observed organonitrates formed in the aqueous phase and they are water soluble.**

R2C7) It is well established that electrospray ionization is highly sensitive to compound structure and the sample matrix. How is it known that the signals assigned to organic nitrates are not just trace components that are not a significant portion of the aerosol mass? Some discussion of quantitation is warranted.

**Response) A standard solution of glyoxal and nitric acid shows dominant organonitrate peaks at m/z⁻ 147 and 226. Therefore, m/z- 147 and 226 in the smog chamber samples cannot be "trace components."**

R2C8) Since the major conclusion of this paper appears to be that organic nitrates can be formed by mixing glyoxal and nitric acid in water in the absence of oxidants, the authors should better explain the point of the environmental chamber experiments. Showing that nitric acid can be formed under these photooxidation conditions seems unimportant, since it is readily formed in the atmosphere. How do the chamber results contribute to the conclusions?

**Response) We added the following in the conclusion section (line 32, page 10):**
**"Our main conclusion is that organonitrates can be formed in wet aerosols during the daytime in the presence of NOₓ in humid areas. Hydrogen peroxide is an OH radical source, and its presence in wet aerosols can be expected when ~ppb is available in the gas phase. HNO₃ formation is facilitated by aqueous photooxidation: NO is effectively converted to NO₂ by volatile organic products (glyoxylic acids) during aqueous photooxidation; and OH generated in wet aerosols by photolysis of hydrogen peroxide evaporates and forms HNO₃ with NO₂. HNO₃ then forms organonitrates with aldehydes and alcohols, dominant water-soluble organic species in wet aerosols. This chemistry is inherently multiphase chemistry."**

Technical Comments
R2C9) Page 1, line 27: "aerosols" should probably be "products"

**Response) "Aerosols" are now changed to "products."**

R2C10) Page 2, line 36: "ethene" should be "acetylene"

**Response) "Ethene" is now changed to "acetylene."**

R2C11) Page 3, line 8: "chromatography" should be "chromatograph"

**Response) "Chromatography" is now changed to "chromatograph."**

R2C12) Page 5, line 26: "nitrate" should be "nitric"

**Response) "Nitrate" is now corrected to "nitric."**

R2C13) Page 9, line 20: Should add "area" after "surface"

**Response) "Surface" is now "surface area."**

R2C14) Page 9, line 38: "catalyzation" should be "catalysis"

**Response) "Catalyzation" is now "catalysis."**

Reference

Smith, D. F., Kiss, A., Leach, F. E., Robinson, E. W., Paša-Tolić, L., and Heeren, R. M. A.: High mass accuracy and high mass resolving power FT-ICR secondary ion mass spectrometry for biological tissue imaging, Anal. Bioanal. Chem., 405, 6069-6076, 10.1007/s00216-013-7048-1, 2013.

---

## Author Response (AR2)

We thank the reviewers for careful reading and helpful comments that improve the quality of the manuscript. Reviewer comments have been copied followed by our responses in bold.

**R1 (Anonymous Referee #2)**

**General Comments**

This paper presents evidence of formation of organonitrates, organosulfates, and mixed nitrate-sulfate organic compounds from glyoxal. Their formation is attributed to aqueous phase processing of glyoxal and its hydrated forms. While photochemistry of gas-phase species enhanced the formation of the organitrogen and organosulfar compounds, the authors present reasonable evidence that this was due to enhanced formation of HNO3, which partitioned into aerosol and enhanced aqueous processing.

Also, the products were formed during photochemical experiments were also formed without UV irradiation. An existing model of aqueous-aerosol glyoxal chemistry was modified to include some new reactions and partitioning of glyoxal. The formation of these interesting organitrogen and organosulfur compounds seems clear, supported by identification with mass spectrometry and a fairly straightforward experimental design.

The data to support conclusions regarding kinetics of these reactions and subsequent modeling is limited, and it is not clear if any strong conclusions can be made by comparison with a kinetics model. The main result of this work is the identification of the products, with the potential of their formation in atmospheric aerosol via the aqueous chemistry presented here.

R1C1) No attempt was made to track the total amount of oxidized forms of nitrogen (e.g. NOx, organonitrates, HNO3), and this should be done during revision. For example, can the observed changes in gas-phase NOx levels be reasonably attributed to known sinks?

Response) We did not attempt to measure HNO3 uptake quantitatively. A FACSIMILE model predicts ~500  $\mu$ M of HNO3 uptake in the aqueous phase after 3 hour irradiation when initial conditions are 10 ppb of NO, 0 ppb of NO2, 10 ppb of O3 and 500 ppb of an organic compound in the gas phase (glyoxylic acid). This concentration of HNO3 is comparable to that of inorganic constituents in wet aerosols (200  $\mu$ M of ammonium sulfate/sulfuric acid). We added this point at the end of section 3.3.

"We also estimate the concentration of HNO3 taken up into wet aerosols by including chemistry of NOx, HOx, peroxy radical, HNO3 partitioning into a FACSIMILE model. ~ 500  $\mu$ M of HNO3 uptake in the aqueous phase is predicted after 3 hours of irradiation when initial conditions are 10 ppb of NO, 0 ppb of NO2, 10 ppb of O3 and 500 ppb of an organic compound in the gas phase (glyoxylic acid). This concentration of HNO3 is sufficient to form organonitrates with glyoxal and comparable to that of other inorganic constituents in wet aerosols (200  $\mu$ M of ammonium sulfate/sulfuric acid)."

R1C2) The lack of detection of glyoxal or its hydrated forms in humidified ammonium sulfate aerosol, even at the beginning of the experiment, is somewhat puzzling and must be explained further.

Response) Thank you for pointing this out. We reinvestigated and found that  $m/z^+ 131$  ([glyoxal +  $2H_2O + Na]^+$ ) was actually detected in 15 and 30 minute samples. But  $m/z^+ 131$  that we detected in 0 minute sample was an imine (C4H7N2O3), which appeared to overlap the glyoxal peak. So, a decay plot cannot be constructed.

We changed the sentence in the paragraph (line 15, page 8) as follows: "(glyoxal was also detected for AS aerosols in the humid chamber; however, since it was only detected in 15 and 30 minute samples, no decay plot was constructed)"

We also added the following in the text (line 12, page 8):

**"131 (= [M + H2O + MeOH + Na]+)"**

This work is significant in the identification of formation of organitrogen and organosulfur compounds from glyoxal chemsitry. Therefore I recommend this work for publication, pending revisions. Fundamental points still need to be addressed, and a number of clarifications are required prior to publication, as detailed below.

**Specific Comments**

R1C3) (2, 39) It should be noted that the low volatility of glyoxal results largely from the high level of hydration that occurs upon dissolution in water.

Response) Clearly glyoxal undergoes hydration, but immediately hydrated glyoxal forms oligomers via acid catalysis and organic-inorganic complexes in the presence of inorganic constituents. These are likely to be SOA from glyoxal. And these points were already addressed in the previous paragraph (line 10-18, page 2).

We have modified that sentence to clarify the reviewer's point: "Water soluble organic compounds like glyoxal and methylglyoxal hydrate and form oligomers through hemiacetal formation and aldol condensation, especially in evaporating droplets."

R1C4) (4, 6) It is has been shown that drying can induce chemistry in aqueous aerosols. (1-3) The aerosol in this study contained glyoxal prior to drying and addition to the chamber. Were there any indications that chemistry occurred during that drying process?

Response) Yes. In Fig. S6, solutions and aerosols in the chamber at 0 minute were directly compared. Imines and acid-catalyzed oligomers were found in the aerosols. This is discussed in the text (line 13, page 10).

R1C5) (4, 15) The humidifying process should be described in further detail. It is currently described as the chamber being filled with clean dry air and then humidified. It doesn't seem feasible that 90% RH can be reached with the chamber initially full of dry air.

Response) This humidifier was developed by modifying an existing commercial humidifier. Water spray and evaporation pan were specially designed to generate water steam rapidly. While adding water vapors into the smog chamber, we monitored SMPS to ensure that no water droplet was introduced. This device will be requested for patent in the future, so we cannot discuss more in details.

**We added:**

"This humidifier was developed by modifying an existing commercial humidifier. Water spray and evaporation pan were specially designed to generate water steam rapidly. While adding water vapors into the chamber, we monitored SMPS to ensure no water droplet was introduced."

R1C6) (4, 12) Please elaborate on the relevance of your gas and particle phase concentrations to the atmosphere. Although the goal of this study is largely to show the potential source of these compounds and the link to aqueous processing, the relation to the atmospheric conditions should be addressed further.

**Response**) We added the following:**

"Concentrations of NOx, O3, and particle mass in smog chamber can be related to a moderate haze condition in urban areas, particularly observed in Seoul or Beijing."

R1C7) (4, 23) The use of E-AIM will also provide, as you note, the pH of the aerosol, yet pH is not reported here. pH will affect particle equilibria, partitioning, and may change the resulting chemistry. It is certainly an important environmental variable that should be reported for all experiments in Table S1. A general comment on pH and potential effects should be included in your updated discussion, particularly since acidity was a major aspect of your experiments (sulfuric acid seed vs. ammonium sulfate seed).

**Response) We have added pH values in Table S1. It appears acidity affects oligomerization in SA aerosols, and this is discussed in line 22-29 on page 10.**

R1C8) (5, 13-19) This section has the heading ": : : and 226", but no mention is made of m/z 226.

**Response) We change the section title to: "MS/MS analysis for $m/z^{-}$ 147 and standard MS analysis for $m/z^{-}$ 147 and 226"**

R1C9) (6, 12-15) The authors observed that experiments that are similar, except for the presence of glyoxal (#2 and #7), had very different NOx chemistry, but do not explain this. The sinks and consequences of gas-phase NOx should be more clearly discussed, particularly in light of your observations. For example, if NOx is converted to HNO3 and partitions to aerosol, pH could be significantly altered.

Response) Please see our response for R1C1 regarding sinks of NOx to HNO3. NO2 is effectively formed by peroxy radical-NO reactions (#2), but is only slightly increased due to the lack of peroxy radicals when there is no glyoxal in the beginning (#7). This point is already discussed in line 10 on page 6. There was no evidence of enhanced oligomer formation by HNO3 uptake while more oligomers were observed in SA aerosols. We modified the text (line 3, page 10):

"However, acidity effects on oligomer formation require further study because sulfuric acid in SA aerosols appears to enhance oligomerization while photochemically formed nitric acid does not."

R1C10) (7, 9-11) Were NO2+ to be formed in any significant amount, would this now be a potentially important reactive species (nucleophile) in your aerosol? Are there any indications that this is the case?

Response) We did not attempt to measure  $NO_2^+$ , so heterogeneous reactions of  $NO_2^+$  is beyond the scope. In this paper, we focus on nitrates and organonitrates. We just mention this since no  $HNO_3$  was observed in SA aerosols.

**We have now added:**

"If  $NO_2^+$  were formed in a significant amount, it could be an important reactive species. However, measurement of  $NO_2^+$  and investigation of its potential role is beyond the scope of this study."

R1C11) (8, 15) The authors state that no glyoxal peak was observed in mass spectra for the humid chamber AS aerosols, yet you do observe organonitrate products (Fig. 1). Does your model suggest complete and rapid conversion of glyoxal to products? Given the importance of ALW for partitioning of glyoxal, it is puzzling that AS aerosol under humid conditions does not contain glyoxal. This important point was dismissed by the authors.

**Response) See our response for R1C2.**

R1C12) (9, 32) The aerosols evaporate to maintain equilibrium at the RH conditions of the chamber, not due to surface area considerations. What comment about surface area was intended?

**Response) This has been changed in the text to:**

**"When the solutions are atomized and introduced into the smog chamber, water evaporates to equilibrate to the chamber RH, and concentration of solutes increase."**

Technical Comments R1C13) (Page 1, Line 13) change to read "or sulfuric acid particles"

**Response) It was already written as "or sulfuric acid particles."**

(Page 1, Line 33) This sentence is awkward, but it makes an important point that SOA(aq) is likely to improve model predictions. Please make this sentence clearer, perhaps split into two.

**Response**) Now it reads:**

"Including SOAaq is likely to improve model predictions, which currently underestimate the ambient concentration and oxidation state of organic aerosols. Water soluble organic compounds with a small carbon number (C2-C3) are not considered precursors to SOA formation through gas-phase chemistry and vapor pressure driven partitioning (Pankow, 1994) because of their high vapor pressure. However, they are potential SOAaq precursors."

R1C14) (2, 12) Add references 4 and 5.

**Response) We add suggested references.**

R1C15) (2,18) Add reference 6.

**Response) We add suggested references.**

R1C16) (2, 20) change to read "..compounds, OH radicals, and water.."

**Response) Now it reads as the reviewer suggests.**

R1C17) (3, 6) The importance of more realistic aerosol composition should be noted here. Ambient aerosol will have a wide range of organic compounds in addition to those derived from glyoxal.(Refs 7,8) Recent work suggests that compounds like glyoxal will from condensation products (acetals, etc.) with these other aerosol constituents.(Ref 9)

This could affect the chemistry studied in this work, by reducing the amount of glyoxal available for reaction and potentially changing the product distribution. The authors should address the effect of actual ambient aerosol composition.

Response) Glyoxal is a surrogate of water soluble organic compounds in ambient wet aerosols. Radical and non-radical reactions of water soluble organic compounds in wet aerosols are expected to be seen in our glyoxal reactions. Glyoxal undergoes self-oligomerization through hemiacetal formation and aldol condensation leading to light absorbing products (Shapiro et al., 2009). Ammoniums, sulfates and nitrates are main inorganic constituents (Zhang et al., 2007), and glyoxal reacts with them. OH reactions of glyoxal produce dicarboxylic acids (e.g., oxalic acid), which are also expected to be the products of water soluble organic compounds. These non-radical reactions compete with OH reactions. In our smog chamber experiments, non-radical reactions are more dominant than OH reactions in the condensed phase and we expect this is true for water soluble organic compounds in ambient wet aerosols.

**We added the following in the sentence:**

"Note that glyoxal is used, as a surrogate of water soluble organic compounds in ambient wet aerosols, to explore non-radical and radical reactions in the condensed phase leading to SOA."

R1C18) (3, 23) remove the first word : "the"

**Response) Now "the" is removed.**

R1C19) (3, 35) change to read "..., liquid water, and : : :"

**Response) We do not understand this request. Perhaps the page or line number the reviewer is referring to is incorrectly written??**

It appears the reviewer indicates (3, 26). We insert the comma between "water" and "and."

R1C20) (5, 30-31) change to read "::::08C11) and not likely nitric acid adducts.."

**Response) Now "and" is inserted**

R1C21) (5, 37) change to read "..Cole, 2000), and MIDAS does not propose.."

**Response) Now "while" is changed to "and."**

R1C22) (6, 12-13) change to read "Experiment #2) also shows significant.."

**Response) Now "the" is removed, as suggested.**

R1C23) (6, 16) This is an interesting style of using an introductory question. It would be better to use a direct statement rather than giving the reader some suspense. Ambiguity impedes clarity. Please rephrase as a direct statement, such as "Aqueous phase chemistry and photochemistry may lead to volatile products that contribute to gas phase peroxy radicals"

**Response)** Now it reads as the review suggests:**

"Photochemistry on wet aerosols may lead to volatile organic products that contribute to gasphase peroxy radicals."

R1C24) (6, 37) and (7, 1) and throughout the manuscript, change to read ": : :after 3 hours of irradiation: ::"

**Response) Now we changed according to the reviewer's suggestion.**

R1C25) (7, 3-5) While heterogeneous reactions are a possible source, do the authors consider OH + NO2 a source of HNO3? Is this included in the model?

Response) We include this as an aqueous phase reaction. NO2 concentration in the aqueous phase is determined by the Henry's law constant. The contribution of this reaction to HNO3 is, however, small.

R1C26) (7, 12) change to read "Figure 2 suggests: : :"

**Response) We removed "also" to read as suggested.**

R1C27) (8, 3-4) change to read "During irradiation, oxalic acid was formed in the humid chamber, shown by UPLC-Q-TOF-MS detection of m/z- 89: : :"

**Response) We changed according to the reviewer's suggestion.**

R1C28) (9,28) through (10, 29) The time-resolved data should be addressed within the context of other studies. Particularly for the reduced nitrogen species (imines, imidizoles, etc.) Studies have looked at this reaction under a wide range of conditions, which should allow comparison.(Refs 10-12)

**Response) We already discussed previous studies of nitrogen-containing organics (line 13-15, page 2). So we have added the suggested references (10 and 11) there. Ref 12 was already cited there.**

R1C29) (10, 18) change to read ": : : because aqueous phase reactions of glyoxal with ammonium form imines..."

**Response) Now we remove "in the" to read as suggested.**

R1C30) (10, 22-23) change to read ": : : form oligomers and imines. In SA aerosols the formation: : :"

**Response) We split the sentence into two as the reviewer suggests.**

R1C31) (11, 8) change to read ": : : during the daytime. Notably, nitrate concentrations: : :" Figures

**Response) Now we removed "And," as suggested**

R1C32) Scheme S1. This should be placed into the main manuscript. You discuss extensively the formation of these organonitrates, so this should not be supplemental.

**Response) We agree. Now Scheme S1 is Scheme 4. The previous Scheme 4 is now Scheme 5.**

R1C33) Figure 1. This figure should be a 4 panel grid, with the spectra for humid conditions on the top row, and dry conditions on the bottom row, with AS results on the left and SA results on the right. It is difficult to compare in a single column. Each figure (a-d) should have a label denoting the aerosol type and the humidity level.

**Response**) Figure 1 is now a 4 panel grid as the reviewer suggests.**

R1C34) Figure 3. The legends must be moved to the top right corner to avoid confusion between the data and the legend. R-squared should be reported to at most 3 decimal places (0.001). The linear fits do not need to fully displayed, only the time constants. The linear fit intercepts should all be 1.0. Instead of presenting the equations, you should label the plots with the effective lifetime or the half-life of the glyoxal.

**Response) Legends are now placed on the top right corner. Y-intercept value (at t = 0) for A is now 1. Instead of the equations, the lifetimes, $\tau$ (min), are used. R-squared has now 3 decimal places. We modify the text (line 18-24, page 8) as follows:**

"For SA aerosols, the lifetime of glyoxal in the dry chamber (51.3 min in Fig. 3A) is very similar to that in the humid chamber (54.1 min in Fig. 3B) due to high hygroscopicity of sulfuric acid (32% ALW in the dry chamber). Assuming no evaporation of ALW, the kinetic model (Details are discussed in the next section) predicts that the lifetime of glyoxal by OH reactions in the aqueous phase is 55.6 min, which is very similar to estimated values above. However, for AS

aerosols in the dry chamber, glyoxal peaks at m/z+ 113, 117 and 131 decay sharply in 30 minutes and the estimated lifetime is 10.9 min (Fig. 3C), which is ~ 5 times shorter than the lifetime by OH reactions."

R1C35) Figure 4. The same 4-panel grid format as suggested for Figure 1 should be used.

**Response**) Figure 4 is now a 4 panel grid as the reviewer suggests.

We thank the reviewers for careful reading and helpful comments that improve the quality of the manuscript. Reviewer comments have been copied followed by our responses in bold.

**R2 (Anonymous Referee #1)**

In this paper the authors describe results of environmental chamber experiments in which they investigate the formation of organic nitrates in aqueous aerosol from reaction of glyoxal with OH radicals under high NOx conditions. Experiments were also conducted in the dark with O3 and probably NO3 radicals and in the absence of oxidants. Aerosol products were collected on filters and analyzed offline by electrospray mass spectrometry to obtain elemental formulas that were used to assign compound identity.

The experiments are well done and the discussion of results is very thorough and reasonable, although I have concerns about proposed product assignments and reaction mechanisms, as noted below. The results have the potential to be important, since organic nitrates are an important class of compounds. Determining the extent to which those present in the atmosphere are formed through aqueous chemistry matters because whereas gas phase formation from RO2 + NO reactions sequester NOx and radicals, aqueous phase formation from aldehydes/alcohols and nitric acid does not. I think the paper may be suitable for publication in ACP but have some significant comments that should first be addressed.

**Specific Comments**

R2C1) Page 4, lines 28-29: It is well known that sonication in water can form hydrogen peroxide and possibly other oxidants. What tests were conducted to ensure that this did not affect the composition of the samples?

Response) In our previous reaction vessel experiments, we used excess hydrogen peroxide as an OH radical source. Addition of hydrogen peroxide to standards verified that hydrogen peroxide does not react with glyoxal or oxalic acid. It does oxidize glyoxylic acid (Tan et al., 2009). However, in this smog chamber experiments, glyoxylic acid evaporates and becomes a major source of peroxy radicals during the irradiation (line 30, page 6). Therefore, a small amount of hydrogen peroxide formed during sonication is very unlikely to affect the sample composition. We add the following:

"Note that any possible hydrogen peroxide formed during sonication is not likely to affect aqueous-phase photooxidation. According to our previous reaction vessel experiments (Tan et al., 2009), hydrogen peroxide does not react with glyoxal or oxalic acid. It only oxidizes glyoxylic acid. But in this smog chamber experiments, glyoxylic acid evaporates."

R2C2) Page 5, line 7: 150 ppm uncertainty in mass assignments seems large. I am used to values less than 5 ppm for accurate assignments of elemental formulas. Why is this, and how does this affect the assignment of elemental formulas? For example, in Tables 1-3 simple molecules like sulfuric acid and its dimer can be identified at the sub- 5 ppm level, but the proposed organic compounds cannot. This seems problematic. This issue should be discussed, and the authors should show what other possible products can be assigned with similar or better uncertainties. As it is, I do not have much confidence in these assignments, including the organic nitrates that are the focus of the paper.

Response) Unlike FTICR-MS, whose uncertainty is below 1 ppm, TOF-MS tends to have uncertainties. Actually this is not surprising because it is well known that uncertainties of TOF could be high (Smith et al., 2013). Note that throughout this work, if MIDAS provided more than one possible chemical formula, we selected the one with the lowest ppm difference. Regarding organonitrates, we also conducted an MS/MS analysis for m/z- 147, which clearly indicated organonitrates by showing a nitrate fragment (m/z- 62) (Fig. S2) and an ESI-MS analysis for a standard mixture solution of glyoxal and nitric acid, confirming m/z- 147 and 226 were organonitrates (Fig. S3). Structures and mechanisms are proposed based on chemical

**formula provided by MIDAS with the lowest ppm difference.**

**We added the following in the sentence:**

"Note that unlike Fourier transform ion cyclotron mass spectrometry (FTICR-MS), whose uncertainty is below 1 ppm, it is well known that uncertainties of TOF could be high (Smith et al., 2013)."

R2C3) Page 5, lines 32-33: It is difficult to believe that an alcohol would be converted to an organic nitrate in aqueous solution by a reversible reaction (the single arrow shown should be a double arrow for equilibrium). I would expect the water to shift the product distribution fully towards the alcohol. If such chemistry can occur it should be known for polyols or at least simple secondary alcohols, so literature should be cited and discussed to support these speculations. Although the mechanism provides an explanation for these particular products, it seems that one would expect many other organic nitrates from this mechanism as well. Why only these two out of many possibilities? This needs more discussion.

Response) We provided references for the formation of organonitrates (nitrate esters). As we all know, alcohols and sulfuric acid form organosulfates (sulfate esters). Esterification for organonitrate and organosulfate formation seems to be more favored than hydrolysis in the condensed phase (aerosol phase). Even in solutions we observed organosulfate formation (m/z-217), as shown in Table 3. Yes, other organonitrates could form through this mechanisms; however, the chemical formula provided is the one with the lowest ppm difference suggested by MIDAS.

**We have provided references, and we now state:**

"The proposed formation and molecular structures are illustrated in Scheme S1. Other organonitrates may form through this mechanism as well. However, the chemical formula provided herein has the smallest error compared to the measured mass."

**We replaced a single arrow by a double arrow.**

R2C4) Page 7, line 33: Should "hydroxide" be "nitrate"? Where would one find ammonium hydroxide in aerosols?

Response) Ammonium hydroxide was used in our previous experiments (Ortiz-Montalvo et al., 2014; Tan et al., 2009). To clarify, we have modified this sentence: "While sulfuric acid and ammonium hydroxide addition do not interfere the real-time formation of oxalic acid in dilute (cloud-relevant) photooxidation experiments..."

R2C5) Page 10, line 14: What is meant by "deprotonated acids"? Do you mean carboxylate ions? Ions do not evaporate from solutions.

**Response) Page 10 should be page 9. We changed "deprotonated acids" to "undissociated acids."**

R2C6) Page 11, lines 5-11: Should be more specific in this paragraph. Do you mean total nitrates or inorganic nitrates or organic nitrates? Organic nitrates are not very soluble in water unless they are low molecular weight and multifunctional.

Response) We mean nitrates. They are major constituents, and form organonitrates in aerosols. They facilitate water uptakes. Alkylnitrates (= organic nitrates or organonitrates) formed in the gas phase were already discussed in line 32-38, page 10. And we observed organonitrates formed in the aqueous phase and they are water soluble.

R2C7) It is well established that electrospray ionization is highly sensitive to compound structure and the sample matrix. How is it known that the signals assigned to organic nitrates are not just trace components that are not a significant portion of the aerosol mass? Some discussion of quantitation is warranted.

**Response) A standard solution of glyoxal and nitric acid shows dominant organonitrate peaks at m/z- 147 and 226. Therefore, m/z- 147 and 226 in the smog chamber samples cannot be "trace components."**

R2C8) Since the major conclusion of this paper appears to be that organic nitrates can be formed by mixing glyoxal and nitric acid in water in the absence of oxidants, the authors should better explain the point of the environmental chamber experiments. Showing that nitric acid can be formed under these photooxidation conditions seems unimportant, since it is readily formed in the atmosphere. How do the chamber results contribute to the conclusions?

Response) We added the following in the conclusion section (line 32, page 10): "Our main conclusion is that organonitrates can be formed in wet aerosols during the daytime in the presence of  $NO_x$  in humid areas. Hydrogen peroxide is an OH radical source, and its presence in wet aerosols can be expected when ~ppb is available in the gas phase. HNO3 formation is facilitated by aqueous photooxidation: NO is effectively converted to  $NO_2$  by volatile organic products (glyoxylic acids) during aqueous photooxidation; and OH generated in wet aerosols by photolysis of hydrogen peroxide evaporates and forms HNO3 with  $NO_2$ . HNO3 then forms organonitrates with aldehydes and alcohols, dominant water-soluble organic species in wet aerosols. This chemistry is inherently multiphase chemistry."

Technical Comments R2C9) Page 1, line 27: "aerosols" should probably be "products"

**Response) "Aerosols" are now changed to "products."**

R2C10) Page 2, line 36: "ethene" should be "acetylene"

**Response) "Ethene" is now changed to "acetylene."**

R2C11) Page 3, line 8: "chromatography" should be "chromatograph"

**Response) "Chromatography" is now changed to "chromatograph."**

R2C12) Page 5, line 26: "nitrate" should be "nitric"

**Response) "Nitrate" is now corrected to "nitric."**

R2C13) Page 9, line 20: Should add "area" after "surface"

**Response) "Surface" is now "surface area."**

R2C14) Page 9, line 38: "catalyzation" should be "catalysis"

**Response) "Catalyzation" is now "catalysis."**

30 chamber.

[revised manuscript text omitted]

| AS (Humid)                                                   | 146.9669
(z = 2) | C6H2N2O12                             | Organonitrate                   | -95.5          |
| · · · · · · · · · · · · · · · · · · ·                        | 154.9581            | C2H3O6S1 Glvcolic acid-sulfate Ester* |                                 | -48.3          |
|                                                              | 197.9096            | C2N1O8S1 Nitrooxy-organosulfate       |                                 | -128.4         |
|                                                              | 225.9279            | C4H1N1O8Cl(35)1                       |                                 | -51.9          |
|                                                              | 227.9426            | C4H1N1O8Cl(37)1                       | Organonitrate                   | 26.0           |
|                                                              | 61.9911             | N1O3                                  | Nitric Acid                     | 44.1           |
|                                                              | 96.9603             | H1O4S1                                | Sulfuric Acid                   | 2.0            |
|                                                              | 146.9665
(z = 2) | C6H2N2O12                             | Organonitrate                   | -98.2          |
|                                                              | 181.9389            | C2N1O7S1                              | C2N1O7S1 Nitrooxy-organosulfate |                |
| SA (Humid)                                                   | 186.9598            | C2H3O8S1                              | Oxalic Acid-Sulfuric Acid       | 23.5           |
|                                                              | 197.9053            | C2N1O8S1                              | Nitrooxy-organosulfate          | -150.1         |
|                                                              | 225.9278            | C4H1N1O8Cl(35)1                       | Organanitrata                   | -52.3          |
|                                                              | 227.9404            | C4H1N1O8Cl(37)1                       | Organomitate                    | 16.4           |
|                                                              | 282.8897            | C6H3O11S1                             | Organosulfate                   | -178.4         |
|                                                              | 288.8996            | C8H1O10S1                             | Organosulfate                   | -103.8         |
|                                                              | 61.9925             | N1O3                                  | Nitric Acid                     | 66.7           |
|                                                              | 96.9614             | H1O4S1                                | H1O4S1 Sulfuric Acid            |                |
| AS (Dry)                                                     | 146.9677
(z = 2) | C6H2N2O12                             | Organonitrate                   | -90.1          |
|                                                              | 171.0956            | C6H11N4O2 Organonitrogen              |                                 | 40.0           |
|                                                              | 173.0084            | C6H5O6 Organic Acid                   |                                 | -4.4           |
|                                                              | 62.0241             | N1O3 Nitric Acid                      |                                 | -10.5          |
|                                                              | 89.0389             | C2H5N2O2 Organonitrogens              |                                 | 36.5           |
| $\mathbf{C} \mathbf{A} \left( \mathbf{D} \mathbf{m} \right)$ | 96.9604             | H1O4S1                                | H1O4S1 Sulfuric Acid            |                |
| SA (Dry)                                                     | 172.9572            | C2H5O7S1                              | Glycolic Acid-Sulfuric Acid     | 109.5          |
|                                                              | 186.9687            | C2H3O8S1                              | 3S1 Oxalic Acid-Sulfuric Acid   |                |
|                                                              | 247.0045            | C4H7O10S1                             | Tartaric Acid-Sulfuric Acid     | 113.2          |

 Table 1: Elemental compositions of organic-inorganic compounds at 180 minute irradiation time (UPLC-Q-TOF-MS negative mode analysis)

\*Glycolic acid-sulfate ester was detected by Galloway et al., 2009

Table 2: Elemental compositions of glyoxal and other organic-inorganic compounds at 180 minuteirradiation time (UPLC-Q-TOF-MS positive mode analysis)

| Aerosols
(Conditions) | $m/z^+$              | Elemental
Composition | Compound                         | Error
(ppm) |  |
|--------------------------|----------------------|--------------------------|----------------------------------|----------------|--|
|                          | 69.0491              | C3H5N2                   | Imidazole*                       | 63.4           |  |
|                          | 107.9732
(z = -2) | C4H1O9Na1                | Organic Peroxide                 | -20.0          |  |
| AS (Humid)               | 109.0698             | C4H10N2Na1               | Imine                            | -35.0          |  |
|                          | 145.0652             | C5H9N2O3 Imidazole*      |                                  | 30.5           |  |
|                          | 149.0299             | C5H6N2O2Na1 Imidazole    |                                  | -72.8          |  |
|                          | 203.1056             | C7H12N6Na1               | Imidazole                        | 19.9           |  |
|                          | 95.0307              | C2H7O4                   | Glyoxal**
(dihydrated)        | 33.5           |  |
| SA (Humid)               | 98.9911              | C2H4O3Na1                | Glyoxal**
(monohydrated)      | -143.1         |  |
|                          | 145.0499             | C4H10O4Na1               | Glyoxal
(hydrated by 2 MeOHs) | 19.1           |  |
|                          | 149.0294             | C6H6O3Na1                | Organic Compound                 | 56.9           |  |
|                          | 69.0528              | C3H5N2                   | Imidazole*                       | 116.9          |  |
|                          | 113.0274             | C3H6O3Na1                | Glyoxal
(hydrated by 1 MeOH)  | 57.4           |  |
| AS (Dry)                 | 117.0098             | C2H4O6Na1                | Glyoxal
(dihydrated)          | -51.5          |  |
|                          | 131.0348             | C3H8O4Na1                | Glyoxal
(hydrated by 1 MeOH)  | 25.3           |  |
|                          | 149.0299             | C5H6N2O2Na1              | Imidazole                        | -33.9          |  |
|                          | 203.0614             | C7H11N2O5                | Imidazole*                       | 23.9           |  |
|                          | 95.0415              | C2H7O4                   | Glyoxal**
(dihydrated)        | 80.1           |  |
| SA (Dry)                 | 98.9894              | C2H4O3Na1                | Glyoxal**
(monohydrated)      | -160.3         |  |
|                          | 131.0065             | C3H8O4Na1                | Glyoxal
(hydrated by 1 MeOH)  | -190.7         |  |
|                          | 149.0213             | C6H6O3Na1                | Organic Compound                 | 2.6            |  |

\*Imidazole detected by Kampf et al., 2012 \*\*Glyoxal appeared at t = 0 min, but disappeared during the irradiation

| Aerosols
(Conditions) | m/z⁻                | Elemental
Composition         | Compound                 | Error
(ppm) |  |
|--------------------------|---------------------|----------------------------------|--------------------------|----------------|--|
|                          | 96.9596             | H1O4S1                           | H1O4S1 Sulfuric Acid     |                |  |
| AS                       | 194.9268            | H3O8S2                           | Sulfuric Acid Dimer      | -3.5           |  |
| (Solution)               | 216.9095            | C2H1O8S2                         | Organosulfate            | -10.8          |  |
|                          | 96.9608             | H1O4S1                           | Sulfuric Acid            | 7.2            |  |
|                          | 216.9142            | C2H1O8S2                         | Organosulfate            | 10.9           |  |
|                          | 275.1671            | C13H23O6                         | Organic Acid Oligomer    | 62.1           |  |
| AS (Dry)                 | 311.1689            | C13H27O8 Organic Acid Oligomer   |                          | -7.2           |  |
|                          | 339.1974            | C15H31O8                         | Organic Acid Oligomer    | -14 9          |  |
|                          | 397.0972            | C14H21O13                        | Organic Acid Oligomer    | -3.9           |  |
|                          | 61.9862             | N1O3                             | Nitric Acid              | -35.0          |  |
|                          | 96.9603             | H1O4S1                           | Sulfuric Acid            | 2.0            |  |
|                          | 146.9671
(z = 2) | C6H2N2O12                        | Organonitrate            | -94.1          |  |
|                          | 181.9377            | C2N107S1                         | Nitrooxy-Organosulfate   | -13.2          |  |
|                          | 197.9200            | C2N108S1                         | Nitrooxy-Organosulfate   | -75.8          |  |
| AS (Humid)               | 209.9507            | C3N108S1                         | Nitrooxy-Organosulfate   | 14.8           |  |
|                          | 225.9276            | C4H1N1O8Cl1                      | Organonitrate            | -52.7          |  |
|                          | 243.9025            | C3O9S2                           | Organosulfate            | 14.7           |  |
|                          | 288.9074            | C4H1O13S1                        | Organosulfate            | 24.0           |  |
|                          | 373.8744            | C6N1O14S2                        | Nitrooxy-Organosulfate   | -5.8           |  |
|                          | 401.9027            | C6N3O16S1                        | Nitrooxy-Organosulfate   | 5.5            |  |
|                          | 486.8823            | C10H3N2O17S2                     | Nitrooxy-Organosulfate   | -11.1          |  |
| C A                      | 96.9614             | H1O4S1                           | Sulfuric Acid            | 13.4           |  |
| SA
(Calatian)         | 194.9283            | H3O8S2                           | Sulfuric Acid Dimer      | 4.2            |  |
| (Solution)               | 216.9104            | C2H1O8S2                         | Organosulfate            | -6.6           |  |
|                          | 96.9611             | H1O4S1                           | Sulfuric Acid            | 10.3           |  |
|                          | 275.1615            | C13H23O6                         | Organic Acid Oligomer    | 41.7           |  |
| SA (Dry)                 | 293.1608            | C13H25O7                         | Organic Acid Oligomer    | 0.8            |  |
|                          | 311.1589            | C13H27O8                         | Organic Acid Oligomer    | -39.3          |  |
|                          | 339.1808            | C13H27O9                         | Organic Acid Oligomer    | -43.5          |  |
|                          | 61.9908             | N1O3                             | Nitric Acid              |                |  |
|                          | 96.9622             | H1O4S1 Sulfuric Acid             |                          | 21.6           |  |
|                          | 146.9669
(z = 2) | C6H2N2O12                        | Organonitrate            | -95.5          |  |
| SA (Humid)               | 209.9499            | C10N2O15S1                       | Nirooxy-Organosulfate    | -7.7           |  |
|                          | 225.9277            | C4H1N1O8Cl1                      | Organonitrate            | -52.7          |  |
|                          | 243.9090            | C6O18S4 Organonitrate            |                          | 41.3           |  |
|                          | 288.9016            | C4H1O11S2                        | Organosulfate            | 17.4           |  |
|                          | 311.1672            | C10H23N4O7                       | Organonitrate            | 32.1           |  |
|                          | 339.1943            | C12H27N4O7                       | C12H27N4O7 Organonitrate |                |  |
|                          | 373.8829            | C6N1O14S2                        | Nitrooxy-organosulfate   | 16.9           |  |
|                          | 401.9040            | C6N3O16S1 Nitrooxy-organosulfate |                          | 8.8            |  |
|                          | 486.8861            | C8H7O18S3                        | Nitrooxy-organosulfate   | 12.5           |  |

Table 3: Elemental compositions of organic-inorganic compounds in dark reactions (UPLC-HR-Q-TOF-MS negative mode analysis)

Scheme 1: Nitric acid formation in the UV (A) and in the dark (B)